# Distribution of alkylamines in surface waters around the Antarctic Peninsula and Weddell Sea

Arianna Rocchi[1,2], Mark F. Fitzsimons[3], Preston Akenga[3], Ana Sotomayor[4], Elisabet L. Sà[1], Queralt Güell-Bujons[1], Magda Vila[1], Yaiza M. Castillo[1], Manuel Dall'Osto[1], Dolors Vaqué[1], Charel Wohl[1,5,6], Rafel Simó[1] and Elisa Berdalet[1]

[1]Department of Marine Biology and Oceanography, Institute of Marine Sciences (ICM), CSIC, Barcelona, E-08003, Spain.
[2]Faculty of Earth Sciences, University of Barcelona, Barcelona, E-08028, Spain.
[3]Biogeochemistry Research Centre, School of Geography, Earth and Environmental Sciences, University of Plymouth, Plymouth, PL4 8AA, UK.
[4]Marine Technology Unit (UTM), CSIC, Pg Marítim de la Barceloneta, 37-49, Barcelona, E-08003, Spain.
[5]Centre of Ocean and Atmospheric Sciences, School of Environmental Sciences, University of East Anglia, Norwich, NR4 7TJ, UK.
[6]National Centre for Atmospheric Science, University of East Anglia, Norwich, NR4 7TJ, UK.

*Correspondence to:* Arianna Rocchi (rocchi@icm.csic.es), Elisa Berdalet (berdalet@icm.csic.es)

**Abstract.** Alkylamines, volatile organic nitrogen compounds with small molecular weight, are present in the surface ocean, participate in the marine biogeochemical nitrogen cycle, atmospheric chemistry and cloud formation. Alkylamines have been detected in polar regions, suggesting that these areas constitute emission hotspots of these compounds. However, knowledge of the sea surface distribution patterns and factors modulating alkylamines remain limited due to their high reactivity and low concentrations, which hamper accurate measurements. We investigated the presence and distribution of alkylamines in seawaters around the Antarctic Peninsula and the northern Weddell Sea during the late austral summer and explored their potential links to marine microbiota. Alkylamines were ubiquitous in all analyzed samples, accounting for ~2 % of the dissolved and particulate organic nitrogen pool. The only particulate form found was trimethylamine (TMA), detected for the first time in Antarctic waters at concentrations of $9.7 \pm 4.6$ nM. We efficiently measured dissolved trimethylamine (TMA, $20.9 \pm 15.2$ nM), dimethylamine (DMA, $32.3 \pm 32.7$ nM) and diethylamine (DEA, $7.2 \pm 1.7$ nM) across the surveyed area, while dissolved monomethylamine (MMA, $12.7 \pm 0.1$ nM) remained below detection limit in most samples. Variations in alkylamine concentrations did not align with the overall phytoplankton biomass but with specific biological components. TMA was predominantly associated with, and released from, nanophytoplankton. DMA was likely produced by the degradation of TMA or trimethylamine oxide by nanophytoplankton cells or associated

heterotrophic bacteria. The sources of DEA remain unclear but were suggestive of a distinct biogeochemical pathway from those of TMA and DMA. MMA is thought to primarily originate from bacterial degradation of nitrogen-based osmolytes or amino acids, but detection in too few samples precluded any robust association with microbiota. This study reveals that volatile alkylamines are widespread in Antarctic surface waters, where they are primarily sourced from nanophytoplankton cells and associated heterotrophic bacteria and protists.

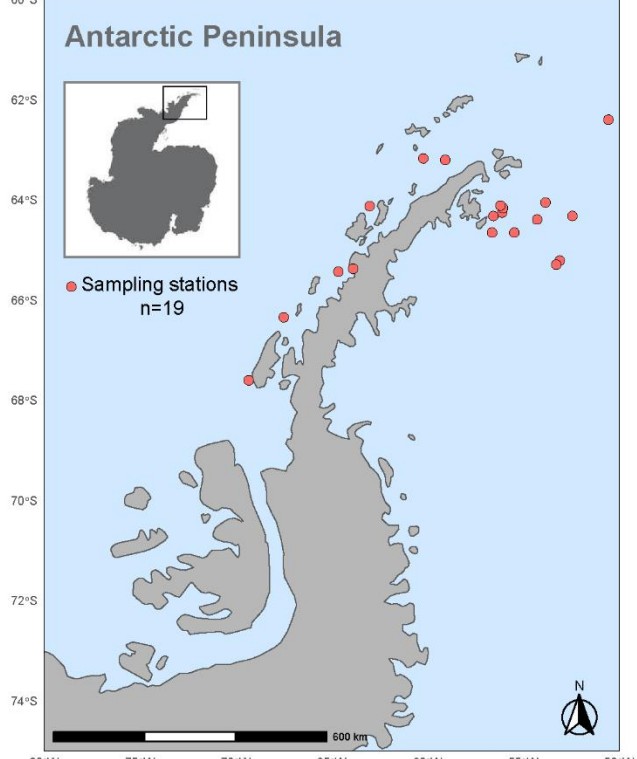 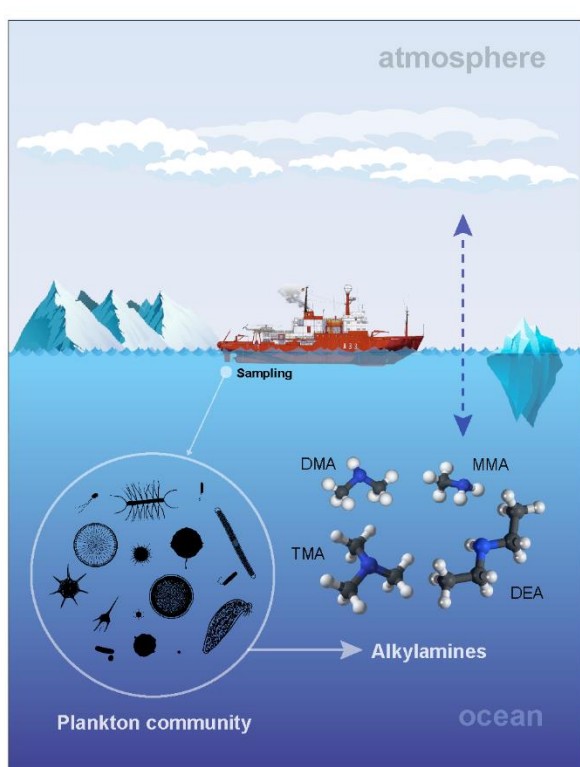

**Short summary.** During the PolarChange expedition, volatile alkylamines, important players in nitrogen cycling and cloud formation, were measured in Antarctic waters using a high-sensitivity method. Trimethylamine was the dominant alkylamine in marine particles, associated with nanophytoplankton. Dissolved dimethylamine likely originated from trimethylamine degradation, while diethylamine sources remain unclear. These findings confirm the biological origin of alkylamines in polar marine microbial food webs.

## 1 Introduction

The marine organic nitrogen (ON) pool is an important natural reservoir of reactive molecules, containing biologically relevant compounds which contribute to biogeochemical cycles in the surface ocean and ocean-atmosphere-climate interactions. Among them, alkylamines are low-molecular weight (<100 Da) polar molecules that exhibit high solubility in seawater and high vapor pressure. Alkylamines are emitted from the ocean to the atmosphere 1) via sea spray, contributing to a highly variable nitrogen-containing fraction of primary aerosols (Dall'Osto et al., 2017; Liu et al., 2022), and 2) through gas exchange, where they are efficiently incorporated into secondary marine aerosols and contribute to very fast new particle formation events (Brean et al., 2021; Corral et al., 2022; Ning et al., 2022; Zu et al., 2024). Additionally, Antarctic sea-ice microbiota and sea-ice-influenced ocean systems are significant sources of dissolved organic nitrogen (DON), including alkylamines, to both the ocean and the atmosphere, with notable release during sea-ice melt (Dall'Osto et al., 2017, 2019; Rinaldi et al., 2020).

Despite recent efforts, the quantification of these species in seawater remains a considerable challenge due to their low concentrations and reactivity (Fitzsimons et al., 2023), which hampers understanding of their concentrations in both dissolved and particulate forms. In the ocean, the main alkylamines reported are the class of methylamines (MAs), which exist in primary (monomethylamine, MMA: $CH_3NH_2$), secondary (dimethylamine, DMA: $(CH_3)_2NH$), and tertiary (trimethylamine, TMA: $(CH_3)_3N$) forms, plus diethylamine (DEA: $(CH_3CH_2)_2NH$), a secondary amine with two ethyl groups bound to the amino nitrogen (N) (Goldwhite, 1964). Amine concentrations in seawater are determined by biogeochemical processes, including production and consumption by marine microorganisms (Gibb et al., 1999). Phytoplankton, other protists and bacteria release N-containing compounds such as proteins, amino acids and several forms of amines (Poste et al., 2014) via organism excretion, cell death or lysis. Some of these compounds are directly synthesized by phytoplankton and used as osmolytes for regulating cellular homeostasis in response to salinity variations (Burg and Ferraris, 2008), and as cryoprotectants (Fitzsimons et al., 2024). The precursors for alkylamines are glycine betaine, choline, trimethylamine N-oxide (TMAO), and quaternary amines ($R_4N^+$). These N- (and Carbon, C) containing molecules are progressively degraded to TMA by bacteria, followed by further degradation into the less methylated compounds, DMA and MMA (Lidbury et al., 2015a, b; Mausz

and Chen, 2019; Sun et al., 2019). This displays similarities to the ocean sulfur cycle of
dimethylsulfoniopropionate and dimethylsulfide (DMSP and DMS, respectively) (Stefels, 2000).
Marine bacteria and archaea can use alkylamines as a source of energy and remineralize the
organic N to ammonium (Landa et al., 2017; Lidbury et al., 2015a; Mausz and Chen, 2019).
The few available studies showed that alkylamines represent a small and highly variable
percentage of marine ON compounds in the ocean (Fitzsimons et al., 2023, 2024 and references
therein). The presence of alkylamines in seawater can have ecological implications, serving as
nutrients (C and N sources) for marine microbiota, thereby influencing primary production and
ecosystem dynamics (Chistoserdova et al., 2009; Palenik and Morel, 1991; Taubert et al., 2017).
For instance, in tropical waters van Pinxteren et al. (2019) found an association between
alkylamines and biological tracers such as chlorophyll-a and fucoxanthin, suggesting that they
were produced by marine diatoms. Furthermore, Koester et al. (2022) hypothesised that the broad
array of N metabolites plays a significant role in the interactions between the diatom *Pseudo-*
*nitzschia* and its bacterial microbiome (particularly *Polaribacter*), thus contributing fundamentally
to the ecophysiology of the diatom. Also, Suleiman et al. (2016) showed that interactions between
diatoms and heterotrophic bacteria may be important for marine amine cycling. Investigations into
the co-occurrence and abundance of proteobacteria, diatoms and MAs in the marine water column
have uncovered interkingdom cross-feeding, underscoring the previously underestimated
significance of MAs in the marine N and C cycles (Stein, 2017). Moreover, MAs share a bacterial
oxidation pathway with the climate-relevant sulfur gas DMS into dimethylsulfoxide (DMSO)
(Lidbury et al., 2016). DEA has been previously found in seawater (Poste et al., 2014; Van
Pinxteren et al., 2012, 2019; Fitzsimons et al., 2024) and marine aerosols (Facchini et al., 2008;
Dall'Osto et al., 2019). However, no information exists on production pathways, potential
biological precursors, or transformation processes in seawater. In summary, the amine cycle in the
ocean is related to several microbial processes, which this study sought to explore further.
Here we aimed to investigate the presence, distribution, and potential sources of alkylamines in
Antarctic waters and to enhance our understanding of how these compounds are linked to polar
microbial ecology. To achieve this, we visited the Southern Ocean near the Antarctic coasts, one
of the most pristine environments on Earth, which is a source of ON (Dall'Osto et al., 2017) and
serves as a proxy for preindustrial marine conditions. Surface waters around the Antarctic

Peninsula were analysed using a sensitive and robust method specifically designed for detecting low molecular weight aliphatic amines. We characterized in detail the biogeochemical properties and microbial composition of the same waters to explore the drivers of alkylamine distribution.

## 2 Methods and Material

### 2.1 Study area and sampling strategy

The PolarChange (Aerosol Emissions in Changing Polar Environments) expedition was conducted on board the RV *Hesperides* in the Southern Ocean around the Antarctic Peninsula, during late austral summer from the 14th of February to the 17th of March 2023. During this cruise, we collected surface seawater samples from the underway water inlet ($\sim$4 m deep) to analyse for amines (dissolved and particulate forms) and accompanying microbiota and biogeochemical parameters. Seven stations were located in the western side of the Antarctic Peninsula, and twelve in the eastern side, within the Weddell Sea area (Fig. 1, Table S1). Seawater was obtained at 18:00 (local time), except for samples #2 and #18, which were collected at 12:00 mid-day. Sea surface water temperature (°C) (SST), salinity and density (sigmaT) were measured by the probe SeaBird SB21 connected to the continuous system and surface solar radiation was measured by a radiometer located in the upper deck (model PRR-800) (PAR; W m$^{-2}$).

### 2.2 Alkylamine sampling and analysis protocol

Seawater was collected into 50 mL propylene tubes (Falcon type), which were completely filled. For dissolved amine analysis, seawater was filtered through a 47 mm GF/F filter (0.7 µm pore size) by gravity (ca. 60 minutes, filtration timing depended on the microbial biomass and particulate matter contained in the sampled water) and directly collected into a new 50 mL propylene tube until completely filled. This procedure minimised headspace as indicated by Akenga and Fitzsimons (2024). This filtered water was preserved with concentrated 37 % HCl (analytical grade) at 1 % (v/v) final concentration. The tube was tightly closed and stored in the dark at 4 °C until analysis. In turn, after filtration, the GF/F filter was stored in a 2 mL eppendorf tube at -80 °C for particulate amine analysis.

### 2.2.1 Analysis of alkylamines in seawater by headspace-based solid-phase microextraction and gas chromatography with Nitrogen-Phosphorus detection

Dissolved and particulate amines in seawater were analysed following Akenga and Fitzsimons (2024). Briefly, the method comprises an online, automated headspace solid-phase microextraction step coupled with gas chromatography and nitrogen-phosphorus detection (HS-SPME-GC-NPD), optimising the method reported by Cree et al. (2018). The new protocol has improved precision, throughput and confidence with advantages in sample collection, storage and transport, particularly from remote environments (Fitzsimons et al., 2023). A sample chromatogram is shown in Fig. S1.

**2.2.2 Reagents and labware**

Methylamine standards, monomethylamine (MMA, 99 %), dimethylamine (DMA, 99 %), trimethylamine (TMA, 98 %) and diethylamine (DEA, 99 %) in hydrochloride form were purchased from Thermo Fisher, UK. Cyclopropylamine (CPA, 99 %), analytical grade HCl (37 %), 10 M NaOH and analytical grade NaCl were from Thermo Fisher, UK. All glassware was soaked for 24 h in Decon solution (2 %, v/v) and rinsed with high-purity water (HPW; 18.2 MΩ cm), then soaked in HCl (10 %, v/v) for 24h, rinsed again with HPW and allowed to dry at room temperature (RT).

**2.2.3 Analysis of dissolved alkylamines**

Dissolved amines, i.e., dMMA, dDMA, dTMA and dDEA stock standard solutions were prepared at 94.8, 59.4, 63.7 and 100 nM, respectively, after an accurate dissolution of their chloride salts in HPW. Stock solutions and working standards were acidified with concentrated HCl at a ratio of 1:1000 v/v (acid:solution). Calibration solutions for dMMA, dDMA and dTMA analyses were prepared in the ranges 9.48–94.8, 5.94–59.4 and 6.37–63.7 nM, respectively and at 10–100 nM for dDEA. Aliquots (10 mL) of the solutions were pipetted into 20 mL autosampler glass vials (cleaned as indicated above) then saturated with NaCl (33 %). CPA was used as an internal standard and was added to each vial at a final concentration of 20 nM. The pH of each standard solution was adjusted to > 13.0 through addition of 10 M NaOH solution (250 µL) and the vials were immediately sealed. At this point, alkylamines were converted to gaseous form and diffused into the headspace, where they were adsorbed into the SPME fibre. Blank samples were prepared with HPW and treated with NaCl and NaOH as described. Samples analyses were conducted ~6

months after collection. From each stored sample, three 10 mL aliquots were distributed in glass
vials and treated analogously to the standards.

**2.2.4 Analysis of particulate alkylamines**

We also measured amines in particulates retained on GF/F filters after seawater filtration (section
2.2). Analyses were conducted ~6 months after sample collection. Prior to extraction, each filter
was placed in a 20 mL autosampler glass vial and allowed to thaw inside the vial (one filter per
vial). Subsequently, we added 250 µL of CPA (20 nM final concentration) as internal standard and
500 µL of 10 M NaOH, to liberate gaseous amines, and the vial was tightly sealed. This treatment
was assumed to volatilize the target analytes into the vial headspace in a manner analogous to
dissolved samples. Particulate amine concentrations were quantified using standard amine
solutions, as described previously.

**2.2.5 SPME and gas chromatography**

Details of the automated method are provided in Akenga and Fitzsimons (2024). Briefly, the
process involved extracting analytes onto an SPME fibre after equilibration in an integrated oven
(60 °C), followed by injection of the SPME fibre into the GC (gas chromatography) system.
Thermal desorption of the analytes occurred in the injector port (250 °C), followed by their
separation and detection on a 60 m CP-Volamine column. Once separated, the analytes were
detected by a nitrogen-phosphorus detector at 300 °C. The total run time lasted 25 minutes. Peak
area data acquisition and processing was performed by Thermochromeleon vs. 7.3 software. The
three MAs and DEA were baseline resolved on the column and separated from CPA. The retention
times of MMA, DMA, TMA, DEA and CPA were 7.2, 8.1, 8.6, 12.0 and 11.3 minutes, respectively
(Fig. S1). An $R^2$ value >0.90 was achieved for the calibration of the four alkylamines. The limits
of detection for MMA, DMA, TMA and DEA, were 9.5, 5.9, 1.1 and 4.3 nM, respectively.
Additionally, the calibration curve for dissolved TMA was used to detect particulate TMA values.

**2.3 Biological Parameters**

**2.3.1 Chlorophyll-a**

Between 200 and 750 mL of seawater were filtered through 25 mm Whatman GF/F glass fibre
filters to estimate the total chlorophyll-a concentration. All filters were stored at -20 °C until
analyses conducted on board the *R/V Hesperides*. Chlorophyll-a (Chl-a) concentrations were
estimated fluorometrically after extraction in 90 % acetone at 4 °C for 24h (Yentsch and Menzel,
1963). Readings were conducted on a Turner 10AU fluorimeter calibrated with pure chlorophyll
extract from spinach (Sigma C5357) using a Beckton-Dickinson spectrophotometer. A
Carbon:Chlorophyll ratio of 50 (Jakobsen and Markager, 2016) was applied to estimate the
phytoplankton biomass in terms of Carbon.
**2.3.2 Viral and bacterial abundance and biomass**
Subsamples (2 mL) were fixed with glutaraldehyde (0.5 % final concentration) for viruses, and
with 1 % paraformaldehyde + 0.05 % glutaraldehyde for bacteria estimations by flow cytometry
(FCM). After 15–30 min in the dark at 4 °C, the fixed samples were flash-frozen in liquid nitrogen
and subsequently stored at -80 °C until analysis. Viral (Brussaard, 2004) and bacterial (Gasol and
Del Giorgio, 2000) abundances were measured in a Cytoflex flow cytometer at the ICM-CSIC
laboratory (up to 5 months after sampling). Samples for viral abundance were thawed and diluted
with TE-buffer (10:1 mM Tris: EDTA), stained with 50x SYBR Green I to a final concentration
of 1 %, heated in a 80 °C bath for 10 min and run at a constant flow rate of 60 µL min$^{-1}$ according
to Brussaard (2004). Viruses were determined in bivariate scatter plots of the green fluorescence
of stained nucleic acids *versus* side scatter. Based on their green fluorescent and side scatter
signals, four distinct virus populations (V1–V4) were identified (Fig. S2). Presumably, V1 and V2
populations are dominated by bacteriophages (Biggs et al., 2021); the V3-V4 fractions by
eukaryotic algal viruses (Evans et al., 2009), and V4 fraction correspond primarily to
Haptophyceae (e.g., *Phaeocystis* spp.) viruses (Brussaard et al., 1999, 2005; Rocchi et al., 2022).
Virus biomass was calculated using the carbon virus content factor of 0.2 fgC virus$^{-1}$ (Suttle, 2005).
Thawed samples for bacterial abundance were stained with 50x SYBR Green I at a final 1 %
concentration and incubated for 5 min in the dark. Based on the flow cytometer side scatter *versus*
green fluorescence (FL1) signatures, high nucleic acid (HNA) from low nucleic acid (LNA)
content bacteria were identified (Gasol and Del Giorgio, 2000) (Fig. S3). Bacterial biomass was
obtained from the carbon-to-volume relationship (Norland, 1993) namely, pgC cell$^{-1}$ = 0.12 x
(V)$^{0.7}$, where V is the bacteria volume cells in µm$^3$. Here, an average cell volume of 0.066 µm$^3$
bacteria$^{-1}$ reported for Antarctic waters (Vaqué et al., 2004) was used.
**2.3.3 Pico- and nanophytoplankton abundance and biomass**
Samples for photosynthetic pico- and nanophytoplankton abundances were collected on 5 mL
cryovials, fixed with glutaraldehyde (1% final concentration) and frozen in liquid nitrogen
following Vaulot et al. (1989). Cells were counted by a CyFlow Cube 8 flow cytometer (Sysmex)
at the ICM-CSIC. Phytoplankton cells were detected with a 488 nm laser beam from their
signatures in a plot of side scatter (SSC) *versus* red fluorescence (FL3), separating the
picophytoplankton size class of 1–2 µm (sphere equivalent diameter, SED), and the
nanophytoplankton size classes with SEDs of 2–7 µm, 7–15 µm, and 15–20 µm (Fig. S4). Within
the nanophytoplankton, Cryptophytes (*Cryptomonas* spp.) were identified by their phycoerythrin
signal in the FL3 vs orange fluorescence (FL2) plots (Marie et al., 2014). Biomasses (µg C L$^{-1}$) of
these cell sizes were measured using the formula, pgC cell$^{-1}$ = 0.216 x (V)$^{0.939}$ (V, cell volume;
Menden-Deuer and Lessard, 2000). The phytoplankton cell volume varied between 1.8 and 63 µm$^3$
cell$^{-1}$.
**2.3.4 Nanoflagellate abundance and biomass**
Abundances of heterotrophic and phototrophic nanoflagellates, including *Phaeocystis*, in the size
fraction 2–20 µm (SED) were determined by epifluorescence microscopy. 30 mL seawater
samples were fixed with glutaraldehyde (1 % final concentration), filtered through 0.6 µm black
(25 mm diameter) polycarbonate filters, and stained with 4,6-diamidino-2-phenylindole (DAPI) at
a final concentration of 5 µg mL$^{-1}$ (Sieracki et al., 1985). Filters were placed on slides and kept
frozen (-20 °C). Microscope cell counts of heterotrophic (HNF) and phototrophic nanoflagellates
(PNF) were estimated by the fluorescence response of the cells after blue light illumination using
an Olympus BX40-102/E at 1000X epifluorescence microscope. PNFs were distinguished by red
fluorescence emitted by photosynthetic plastid structures, while HNF were identified from the
yellow fluorescence of DAPI stained nuclei. At least 50 HNFs and 50 PNFs were counted per
sample (3 transects of 5 mm in each filter) and classified into ≤ 2 µm, 2–5 µm, 5–10 µm, and 10–
20 µm size (SED) classes. The nanoflagellate carbon cell content was estimated from the
corresponding carbon-to-volume ratio, e.g., pgC cell$^{-1}$= 0.216 x (V)$^{0.939}$ (Menden-Deuer and
Lessard, 2000), where the cell volume (V) was calculated from the average length of each
nanoflagellate cell size class and transformed into spherical or ellipsoidal volume. The
nanoflagellate cell volume varied between 1.8 and 57.6 µm$^3$ cell$^{-1}$.

### 2.3.5 Microplankton assemblages

The microplankton community was characterised using the Utermöhl method on 125 mL neutral lugol fixed seawater samples. 50 mL aliquots samples were settled in sedimentation chambers for 24 h and observed in a Leica MDi1 inverted microscope (Edler and Elbrächter, 2010). The identified taxa and size classes included: dinoflagellates (resting cysts, vegetative dinoflagellates 10–20 μm, 20–40 μm, and > 40 μm), diatoms (10–20 μm, 20–40 μm, and > 40 μm) and ciliates. When possible, taxa were identified at the genus and species level. The relative biomasses (in μgC $L^{-1}$) were measured from cell volumes using the carbon-to-volume relationships estimated by Menden-Deuer and Lessard (2000) on diatoms and dinoflagellates. Namely, the equation pgC $cell^{-1}$ = 0.760 x $(μm^3 \ cell^{-1})^{0.819}$ was used for dinoflagellates and pgC $cell^{-1}$ = 0.288 x $(μm^3 \ cell^{-1})^{0.811}$ for diatoms. The biovolume was estimated considering an ovoid, cylinder or prism shape for dinoflagellates, centric diatoms, and pennate diatoms, respectively, and empty sphere for empty dinoflagellate cysts. Cell dimensions measurements (excluding chaetae and other cell expansions) were conducted using a digital camera and specific calibration of the used Leica DMi1 microscope. Empty diatom frustules were assumed to have a null contribution to C.

### 2.3.6 Photosynthetic efficiency

The relative efficiency of excitation energy captured by the photosystem II (PSII), calculated as $F_v'/F_m'$, is used as a proxy of phytoplankton stress and fitness (Gorbunov et al., 2020; Gorbunov and Falkowski, 2022). The metric is measured by a multi-color fluorescence induction and relaxation instrument (mini-FIRe system) (Gorbunov et al., 2020). The instrument records two parameters: $F_0'$ as the minimal yield of fluorescence before fast light flashes, and $F_m'$, the maximum yield of fluorescence due to the reradiation of the maximum number of photons. The difference between $F_m'$ and $F_0'$ is called variable fluorescence ($F_v'$). The quotient of $F_v'/F_m'$ represents the effective photosynthetic efficiency of the community measured under light conditions (Gorbunov and Falkowski, 2022). $F_v'/F_m'$ has no units, so that it is independent of the phytoplankton abundance and allows comparisons between environments. Aliquots of 10 mL were sampled from the underway system and rapidly placed in the chamber of the mini-FIRe to apply the induction and relaxation protocol for dilute samples. No dark acclimation period was used.

### 2.4 Chemical parameters

### 2.4.1 Particulate Organic Carbon and Nitrogen

Particulate organic carbon (POC) and nitrogen (PON) content in the seawater was determined by filtration of 390 to 1000 mL through pre combusted (450 °C, 4h) 25mm GF/F glass fibre filters (Whatman) at low pressure (<20mmHg) and kept frozen (-80 °C) until analysis. Filters were thawed and dried at RT, exposed to 37 % (pure) HCl atmosphere in a hermetic beaker to eliminate carbonate salts and subsequently analysed with an Elemental Analyser (Perkin-Elmer 2400 CHN) at the Scientific and Technical Service of the University of Barcelona. In the following, the term POC and PON will refer to the C and N estimated biochemically as described here as a proxy of particulate organic matter, consisting in living and non-living cells, extracellular material and detritus containing C or N.

### 2.4.2 Dissolved Carbon and Nitrogen

For total organic carbon (TOC) and nitrogen (TN: organic and inorganic nitrogen) analyses, 30 mL of seawater was filtered through a HCl clean 200 µm mesh by gravity and collected in polycarbonate bottles. The sample was fixed with 100 µl of 25 % $H_3PO_4$ stored frozen (-20 °C) until analysis in the laboratory. Following the elimination of inorganic C (i.e., carbonates) by the acidification of the sample, determination of TOC and TN in seawater was conducted by high temperature catalytic oxidation (680 °C and 720 °C, respectively) as described in Álvarez-Salgado and Miller (1998). Measurements were conducted with the TOC-L Shimadzu autoanalyzer, with deep Sargasso Sea water used as control (Hansell Laboratory, University of Miami, RSMAS). Concentrations are expressed as µM (µmol C $L^{-1}$ or µmol N $L^{-1}$). Dissolved Organic Carbon (DOC) and Nitrogen (DON) were calculated by subtraction of POC from TOC, and nitrate, nitrite, ammonium and PON concentrations from TN, respectively.

### 2.4.3 Dissolved inorganic nutrient analysis and total Phosphorus

For measurements of nutrient concentrations, seawater samples were collected in two different 50 mL polypropylene plastic tubes: one tube was used for the determination of inorganic nutrients (nitrate, nitrite, ammonium, phosphate and silicate) and the other one for total phosphorus (TP, organic and inorganic forms). Samples were immediately frozen and stored at -20 °C until analysis. Concentrations of inorganic nutrients were determined with an AA3 HR autoanalyzer (Seal

Analytical) and TP with an AA3 autoanalyzer after previous digestion, following Grasshoff et al. (1983).

**2.4.4 Total dimethylsulfoniopropionate (DMSP) concentrations**

Samples for total DMSP (DMSPt) analysis were collected directly from the underway system on ~30 mL borosilicate serum vials and processed following Kinsey and Kieber (2016). The vials were uncapped and individually heated by microwave until they began to boil. After the first bubble formed, the microwave was stopped and the vial was left to cool. Subsequently, 30 µl of 37 % HCl were added to all the vials to remove the DMS present and preserve the DMSP. Acidified samples were stored at RT in the dark. Within six months of the cruise, DMSP was converted to DMS by alkaline hydrolysis with NaOH for at least 24 hours. The resulting DMS was quantified with a cryogenic purge-and-trap system coupled to a Thermo Fisher TRACE 1300 gas chromatograph with flame photometric detection following Masdeu-Navarro et al. (2022).

**2.4.5 DMS measurements by Proton Transfer Reaction Time-of-Flight Mass Spectrometry (PTR-ToF-MS)**

A Vocus-PTR-ToF-MS coupled to a segmented flow coil equilibrator was used to continuously measure seawater dissolved DMS (Wohl et al., 2019). An overview on operation and calibrations is provided in Wohl et al. (2024).

**2.5 Statistical analyses**

All analyses were conducted in the RStudio integrated development environment (RStudio Team, 2023) to ensure reproducibility and clarity. Multivariate statistical analyses were performed using R version 4.3.2 (R Core Team, 2023) to explore relationships among variables. The data were normalised by centering and scaling to ensure equal contribution of all variables to the Principal Component Analysis (PCA). The PCA was conducted to reduce dimensionality and examine the relationships among variables. The analysis employed the princomp() function from the stats package (Bolar, 2019), using the correlation matrix of normalized data as input to focus on inter-variable relationships. Visualizations were generated using the factoextra package version 1.0.7 (Kassambara and Mundt, 2020). The ggcorrplot package (Kassambara, 2021) was used to create a heatmap of variable correlations, while the gridExtra package (Auguie, 2017) facilitated side-by-

side comparisons of variable contributions to principal components. Factor Analysis was
performed to uncover latent structures within the dataset using the psych package version 2.3.6
(Revelle, 2023). Factor extraction employed Principal Axis Factoring with Varimax rotation to
achieve interpretability, complemented by Maximum Likelihood Estimation for comparison.
Factor loadings were visualized using ggplot2 version 3.4.4 (Wickham, 2023). Mantel Test was
used to assess the correlation between two distance matrices using the vegan package version 2.6-
4 (Oksanen, 2022). For each pair of variables, Euclidean distance matrices were computed and
tested for significant Pearson correlations. Results with p-values < 0.05 were considered
significant. The Wilcoxon test and ggplot2 were used to analyze and visualize statistical
differences between the Antarctic Peninsula and Weddell Sea groups, with a logarithmic y-axis
improving data interpretation.

## 3 Results

### 3.1 Cruise setting

The regional satellite images of SST and Chlorophyll concentration during the cruise period (Fig.
1) indicates two distinct areas where the PolarChange cruise was conducted: the Western Antarctic
Peninsula and the northern Weddell Sea. For this reason, in the following we will explore potential
differences between these two areas concerning biological and biochemical parameters (Fig. S5).
Sea surface temperature (SST) ranged between -0.7 and 2.4 °C (Table S1) with statistical
differences within the two studied marine areas, (average ± SD values) 1.9 ± 0.6 °C (n=7) in the
western part of the Antarctic Peninsula compared to the colder waters of the Weddell Sea with 0.2
± 0.7 °C (n=12; p=0.0072) (Table S1 and Fig. S5). Salinity (Table S1) remained relatively constant
throughout the expedition, averaging 33.9 ± 0.3. Concerning solar irradiance (Table S1), higher
but not significantly different values were observed near the Antarctic Peninsula, 355 ± 257 W m$^{-}$
$^{2}$, compared to the 226 ± 194 W m$^{-2}$ numbers observed in the Weddell Sea.

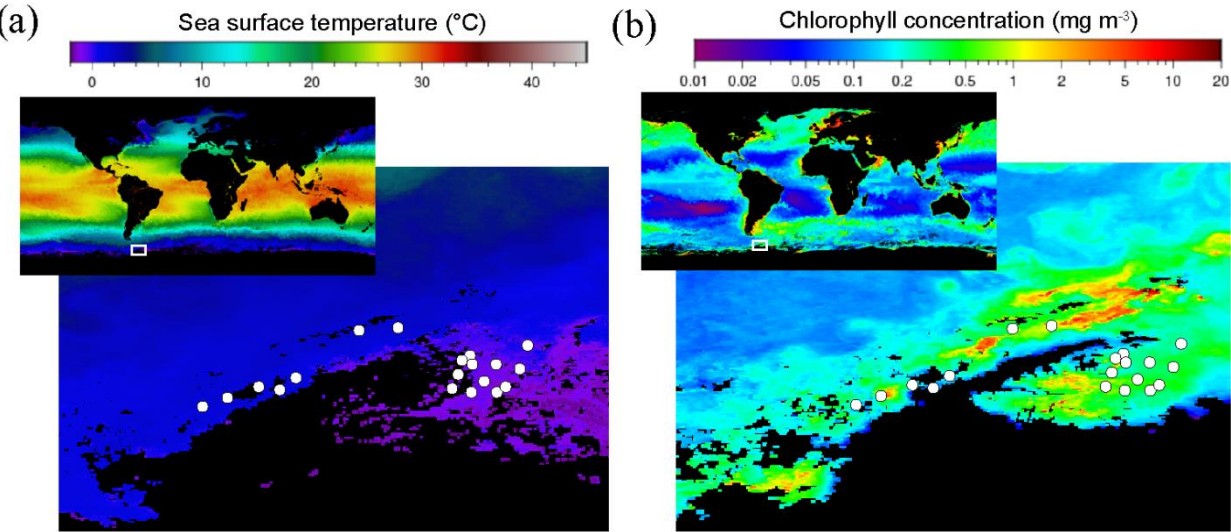

**Figure 1.** Satellite images of (a) the sea surface temperature and (b) the chlorophyll distribution in the ocean (small upper insert) with a zoom in the Southern Ocean around the Antarctic Peninsula and the Weddell Sea in March 2023 during the period of the PolarChange cruise when most amine samples were collected. White circles indicate the location of the 19 stations where all samples analysed in this study were collected (the first seven stations are located in the Western Antarctic Peninsula, while the remaining twelve stations are situated in the Weddell Sea; see stations list in Table S1). Chlorophyll concentration is estimated from the Ocean Color Index (OCI) Algorithm and the sea surface temperature from SNPP VIIRS satellite, https://oceancolor.gsfc.nasa.gov/l3/.

## 3.2 Alkylamine concentrations

### 3.2.1 Dissolved alkylamines

We detected dissolved MAs and DEA at ~4 m of depth over the cruise (Fig. 2a and Table S1). Dissolved MMA (dMMA) was quantitatively identified only in samples #9, #10, #11 in the Weddell Sea with an overall concentration average of $12.7 \pm 0.1$ nM (n=3). With this method we could detect dDMA in most of the samples, ranging from 7.6 nM to 132.3 nM with an average of $32.3 \pm 32.7$ nM (n=15); it was below detection limits in samples #12, #14, #15, #16. The concentration of dDMA was statistically higher near the Antarctic Peninsula compared to the Weddell Sea (respectively, $49.9 \pm 39.6$ nM, n=7 and $17.0 \pm 11.4$ nM, n=8; p=0.04) (Fig. S5). dTMA was measured in all the samples varying from 1.5 nM to 67.9 nM with an average of 20.9 $\pm 15.2$ nM (n=19) ($20.8 \pm 10.6$ nM, n=7 for the Western Antarctic Peninsula and $21.0 \pm 17.3$ nM, n=12 for the Weddell Sea; p=0.77). dDEA was identified in all the samples but with lower concentrations than the dissolved MAs along the studied area (Table S1). dDEA concentrations ranged between 5.1 nM and 13.3 nM, with an average of $7.2 \pm 1.7$ nM (n=19) ($7.7 \pm 2.5$ nM, n=7

for the Western Antarctic Peninsula and 6.9 ± 1.0 nM, n=12 for the Weddell Sea; p=0.77). In this
study, dDEA had the most even distribution of all alkylamines (excluding dMMA), with a
coefficient of variation of 23 %, compared to 101 % for dDMA and 73 % for dTMA.

**3.2.2 Particulate alkylamines**

Only pTMA was detected and identified (Fig. 2b and Table S1) in 18 filter samples (sample #3
was lost), i.e., associated with particles. pTMA showed concentrations ranging between 9.7 nM
and 28.1 nM with an average of 14.4 ± 4.6 nM (14.5 ± 6.2 nM, n=6 for the Western Antarctic
Peninsula and 14.4 ± 3.6 nM, n=12 for the Weddell Sea; p=0.62).

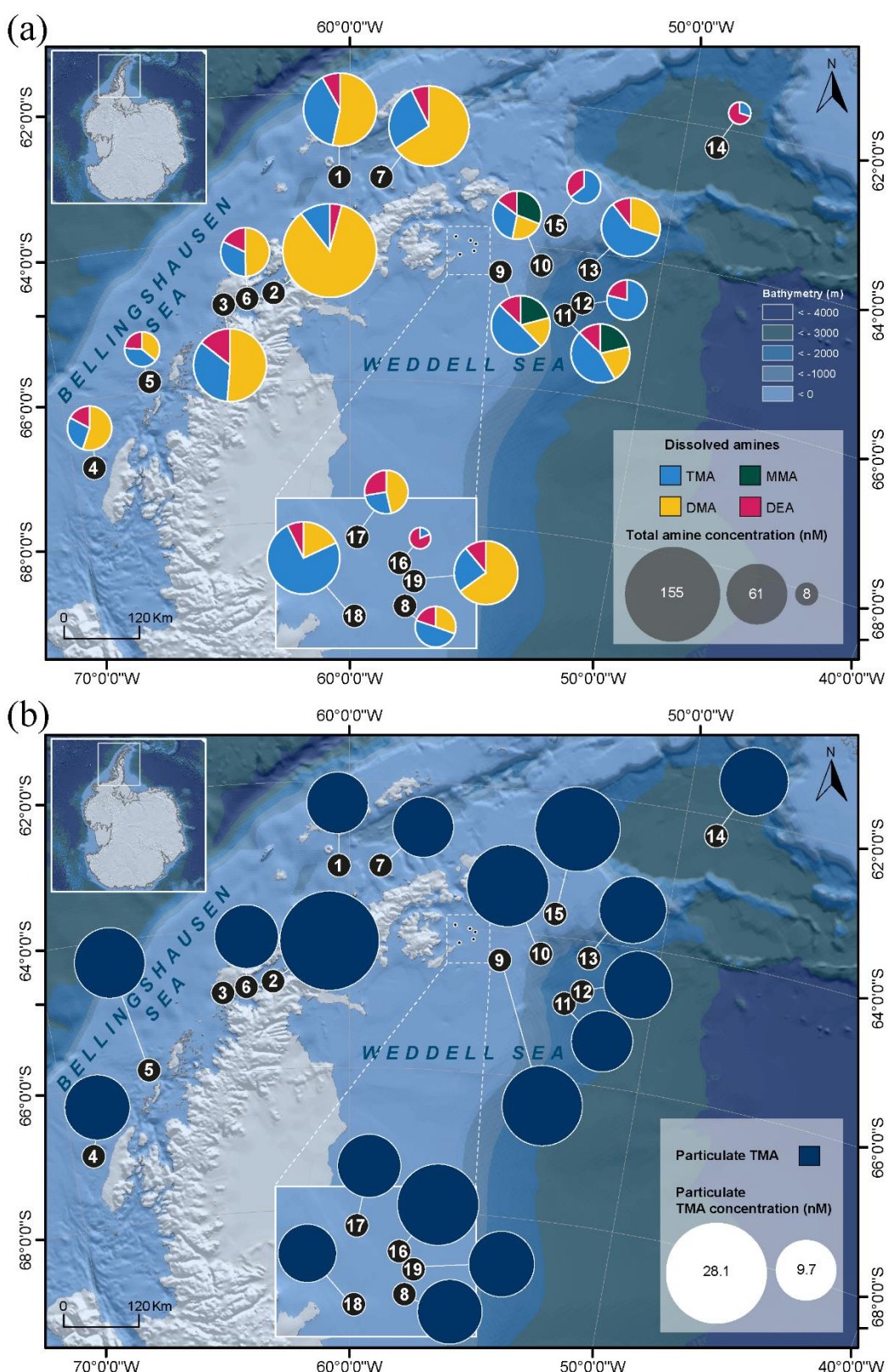


**Figure 2**. Distribution of the concentrations (using pie charts) of (a) the four dissolved alkylamines (MMA, DMA, TMA and DEA) and (b) particulate TMA (*note that the particulate sample #3 was lost) in the studied area.

### 3.3 Biological variables

### 3.3.1 Chlorophyll-a concentrations

The Chl-a concentrations varied along the oceanographic cruise, from 0.2 to 9.6 µg L$^{-1}$, throughout the area (Fig. 1), with an average of 1.2 ± 2.0 µg L$^{-1}$ (n=19) (Table S2). More productive waters were found in the western side of the Antarctic Peninsula, with an average of 2.5 ± 2.9 µg L$^{-1}$ (n=7) significantly higher than the values observed in the Weddell Sea samples (0.5 ± 0.3 µg L$^{-1}$, n=12; p=0.0077) (Fig. S5).

### 3.3.2 Viral and bacterial abundances

Viral abundances (VA) (Table S2) averaged 8.2 ± 3.8 × 10$^6$ viruses mL$^{-1}$ (n=19) and the V1, V2 and V3 populations accounted, on average and respectively, for the 80 %, 16.5 % and 3.5 % of total VA. V4 was only present in sample #15 with an abundance of 1.8 × 10$^5$ viruses mL$^{-1}$. On average, total VA was significantly higher near the Antarctic Peninsula (11.5 ± 3.8 × 10$^6$ viruses mL$^{-1}$, n=7) than in the Weddell Sea (6.2 ± 1.9 × 10$^6$ viruses mL$^{-1}$, n=12; p=0.013) (Table S2 and Fig. S5). V1 abundance was also significantly higher in the Antarctic Peninsula (9.4 ± 3.1 × 10$^6$ viruses mL$^{-1}$, n=7) than in the Weddell Sea (4.9 ± 1.7 × 10$^6$ viruses mL$^{-1}$, n=12; p=0.0098) (Table S2 and Fig. S5). Concerning bacterial abundances (BA), the total average was 6.4 ± 2.5 × 10$^5$ cells mL$^{-1}$ (n=19) with slightly (but not significantly different) higher numbers in the waters near the Antarctic Peninsula (7.0 ± 1.6 × 10$^5$ cells mL$^{-1}$, n=7) compared to Weddell Sea (6.0 ± 2.8 × 10$^5$ cells mL$^{-1}$, n=12). However, the highest value was estimated in sample #16 (11.7 × 10$^5$ cells mL$^{-1}$) (Table S2) collected in the Weddell Sea. Generally, most bacteria had a high nucleic acid content, indicating that more than half of the total bacteria numbers were active cells (Table S2). Note that here, we are referring to cell abundances and not to biomass; C concentration values calculated from cell numbers followed the same patterns as cell abundances for each microorganism described (data not shown in the text, see SI).

### 3.3.3 Pico- and nanophytoplankton abundances

Regarding phytoplankton measured by FCM, the abundances of the five identified groups (1–2 µm, 2–7 µm, 7–15 µm, 15–20 µm and Cryptophytes) were 1.6 ± 1.7 × 10$^3$, 1.8 ± 0.6 × 10$^3$, 5.7 ± 7.5 × 10$^2$, 1.3 ± 2.5 × 10$^2$, 1.5 ± 2.5 × 10$^2$ cells mL$^{-1}$, respectively (average ± SD values, n=19; Table S2). Picophytoplankton cells, ranging from 1 to 2 µm in size, exhibited significantly higher

abundances around the Antarctic Peninsula, with an average of $3.3 \pm 1.8 \times 10^3$ cells mL$^{-1}$ (n=7),
compared to the Weddell Sea ($6.1 \pm 4.1 \times 10^2$ cells mL$^{-1}$, n=12; p<0.001) (Fig. S5). Conversely,
the average abundance of the larger cells, nanophytoplankton, ranging from 2 to 20 µm, appeared
marginally higher in the Weddell Sea ($2.7 \pm 0.9 \times 10^3$ cells mL$^{-1}$, n=12) than in the western part of
the Antarctic Peninsula ($2.2 \pm 1.5 \times 10^3$ cells mL$^{-1}$, n=7). Specifically, the abundance of
nanophytoplankton cells 2–7 µm in size was significantly greater in the Weddell Sea compared to
the Antarctic Peninsula coasts ($2.1 \pm 0.5$ and $1.3 \pm 0.5 \times 10^3$ cells mL$^{-1}$, n=19; p=0.0072) (Fig. S5).
Similarly, cryptophytes (*Cryptomonas* spp.) presented abundances of $112 \pm 143$ cells mL$^{-1}$ (n=7)
in the Western Antarctic Peninsula in contrast to $146 \pm 121$ cells mL$^{-1}$ (n=12) in the Weddell Sea.
**3.3.4 Nanoflagellate abundances**
Abundances of HNF and PNF measured by epifluorescence microscopy were, on average, of 986
$\pm$ 951 cells mL$^{-1}$ and $5046 \pm 2538$ cells mL$^{-1}$ (n=15; samples #5, #9, #11 and #15 were lost),
respectively (Fig. 3 and Table S3). In the Western Antarctic Peninsula, the abundances were 1234
$\pm$ 1195 cells mL$^{-1}$ for HNF and $4240 \pm 1688$ cells mL$^{-1}$ for PNF (n=6). In comparison, in the
Weddell Sea, the abundances were $820 \pm 698$ cells mL$^{-1}$ for HNF and $5583 \pm 2849$ cells mL$^{-1}$
(n=9) for PNF. Concerning size, in the case of HNF, the 2 to 5 µm group, constitutes the largest
proportion of total abundance followed by the smallest size category ($\leq$ 2 µm), the 5 to 10 µm
group, and finally, the largest category ranging from 10 to 20 µm. Similarly, for PNF, the smallest
size categories ($\leq$ 2 µm and 2–5 µm) were the most abundant, followed by the 5–10 µm category,
and lastly, the largest category spanning 10 to 20 µm (Table S3). PNF 5–10 µm showed a statistical
difference between the two Antarctic areas with barely higher concentrations in the Weddell Sea
($117.3 \pm 88.3$ and $193.6 \pm 74.4$ cells mL$^{-1}$, n=15; p=0.045) (Fig. S5). Total PNF exhibited slightly
greater abundances in the Weddell Sea. Additionally, *Phaeocystis* presented slightly lower
abundances west of the Antarctic peninsula of $208 \pm 169$ cells mL$^{-1}$ (n=6) in contrast to $352 \pm 383$
cells mL$^{-1}$ (n=9) in the Weddell Sea (Table S3).
**3.3.5 Composition and abundance of microplankton assemblages**
A diverse range of phytoplankton taxa was found in the studied period in the Antarctic marine
environments (Fig. 3 and Table S4). In the smallest size of the dinoflagellate group (10–20 µm),
the identified taxa were *Gymnodinium* spp., Kareniaceae, *Oxytoxum* spp. and *Prorocentrum*
*cordatum (= P. minimum)*. The intermediate size group (20–40 µm) included larger taxa such as
*Gymnodinium* spp., *Protoperidinium bipes*, *Gyrodinium* spp., Kareniaceae cells, and
*Lebouridinium glaucum (=Katodinium glaucum)*. In the >40 μm category, only *Gyrodinium* spp.
and *Gymnodinium* spp. heterotrophs were present. Among diatoms, in the 10–20 μm size group,
we identified a variety of genera, including centric and pennate chains, *Thalassiosira*, *Porosira*,
*Coscinodiscus*, *Fragilaria*, *Chaetoceros* and *Amphora*. In the 20–40 μm size range, larger cells of
*Coscinodiscus*, *Corethron criophilum* and its spores, pennate chains like *Pseudo-nitzschia*,
*Proboscia alata*, *Licmophora*, *Achnanthes, Navicula*, *Leptocylindrus*, and *Actinocyclus* were
observed. Among the larger diatoms (>40 μm), we identified *Coscinodiscus*, *Corethron*
*criophilum*, and *Chaetoceros* spp.*, Proboscia alata, Lioloma* chains, *Rhizosolenia curvata*,
*Actinocyclus* and pennate diatoms. Non-photosynthetic taxa included mainly tintinnid ciliates.
Dinoflagellates were particularly dominant, though in general, they were distributed close to the
Antarctic Peninsula. Specifically, dinoflagellate cysts accounted for ca. $1.2 \pm 1.1$ x $10^3$ cells $L^{-1}$
(n=7), compared to $0.8 \pm 1.6$ x $10^3$ cells $L^{-1}$ in the samples from the Weddell Sea (n=12).
Dinoflagellates 10–20 μm were found at concentrations of $6.9 \pm 5.8$ x $10^3$ cells $L^{-1}$ (n=7) near the
Antarctic Peninsula, compared to $1.3 \pm 1.2$ x $10^4$ cells $L^{-1}$ (n=12) in the Weddell Sea. Intermediate-
sized dinoflagellates (20–40 μm) had similar abundances in both seas, with $9.7 \pm 5.1$ x $10^3$ cells
$L^{-1}$ in the Antarctic Peninsula waters (n=7) and $1.7 \pm 2.3$ x $10^4$ cells $L^{-1}$ in the Weddell Sea (n=12).
Larger dinoflagellates (>40 μm) were more concentrated in the Antarctic Peninsula waters, with
$1.2 \pm 1.4$ x $10^3$ cells $L^{-1}$ (n=7) compared to $3.2 \pm 4.9$ x $10^2$ cells $L^{-1}$ (n=12) in the Weddell Sea.
Similarly, diatoms were more abundant near the Antarctic Peninsula waters: smaller diatom cells
(10–20 μm) were significantly more prevalent in this area ($2.0 \pm 3.7$ x $10^5$ cells $L^{-1}$, n=7) compared
to the Weddell Sea ($4.7 \pm 9.1$ x $10^5$ cells $L^{-1}$, n=12; p=0.0087) (Fig. S5). Furthermore, sample #1
exhibited the highest abundance of diatoms within the 10–40 μm size range compared to all other
samples (Fig. 3). Intermediate-sized diatoms followed a similar pattern, with $1.2 \pm 2.9$ x $10^5$ cells
$L^{-1}$ (n=7) near the Antarctic Peninsula waters and $6.7 \pm 8.5$ x $10^2$ cells $L^{-1}$ (n=12) in the Weddell
Sea. Larger diatoms (>40 μm) presented significantly higher concentrations ($3.5 \pm 2.9$ x $10^3$ cells
$L^{-1}$, n=7) in the Antarctic Peninsula area than ($8.0 \pm 5.8$ x $10^2$ cells $L^{-1}$, n=12; p=0.028) in the
Weddell Sea (Fig. S5). In contrast, ciliates showed slightly higher abundances in the Weddell Sea,
averaging $4.5 \pm 8.2$ x $10^2$ cells $L^{-1}$ (n=12) compared to $4.1 \pm 3.5$ x $10^1$ cells $L^{-1}$ (n=7) in the Western
Antarctic Peninsula.

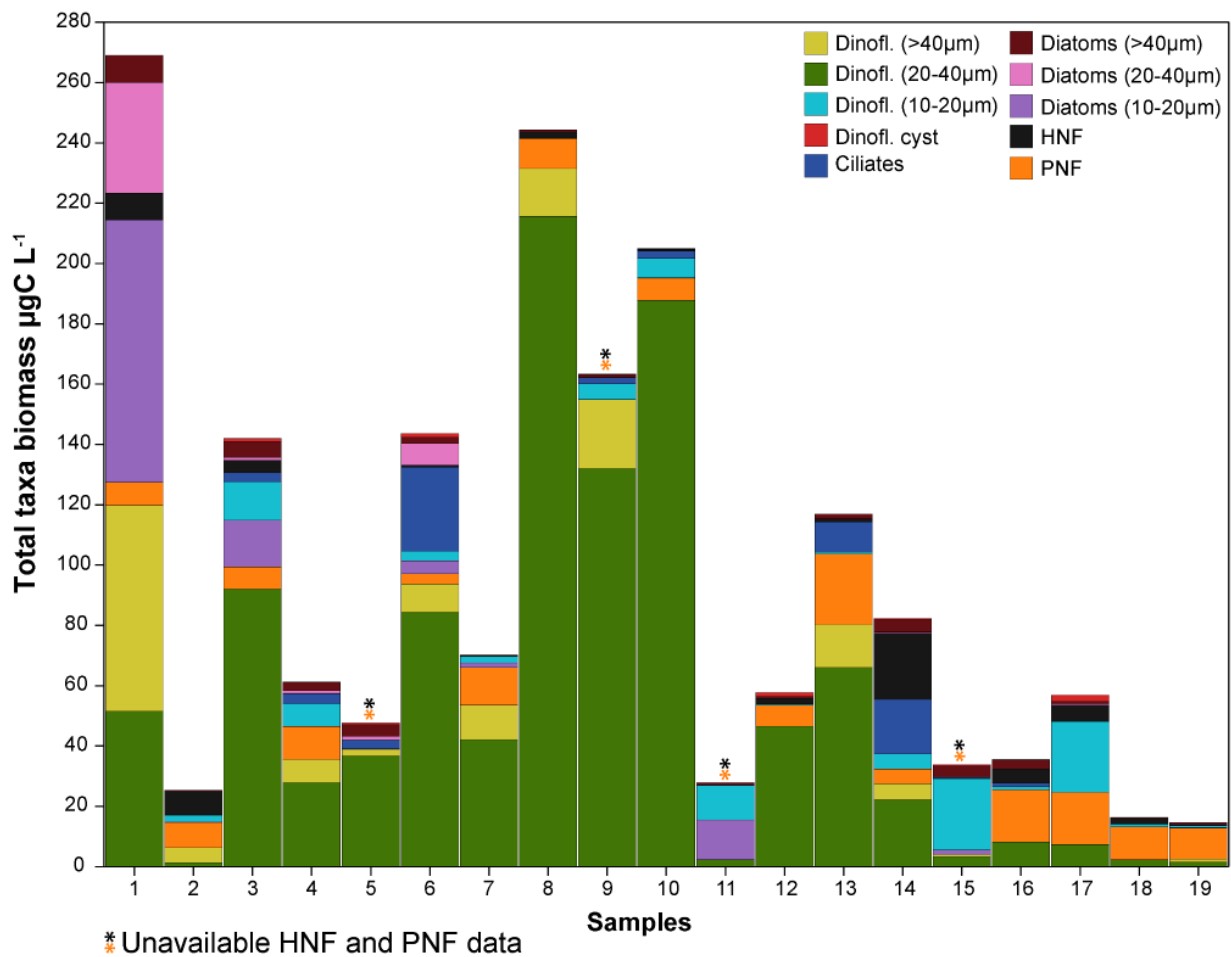


**Figure 3**. Biomass (µg C L$^{-1}$) and proportions (represented by the barplots) of the main phytoplankton groups, protist
and microzooplankton in the 19 samples obtained in the studied area (*note that samples #5, #9, #11, #15 of HNF and
PNF were lost).

### 3.3.6. Photosynthetic efficiency ($F_v$'/$F_m$')

The ecophysiological state and fitness of phytoplankton ($F_v$'/$F_m$') ranged between 0.21 and 0.54,
with an average of $0.38 \pm 0.10$ (n=19). Values were slightly yet not significantly higher in the
samples near the Antarctic Peninsula ($0.41 \pm 0.06$, n=7) compared to the samples collected in the
Weddell Sea ($0.36 \pm 0.11$, n=12; p=0.36).

### 3.4 Chemical variables

### 3.4.1 Organic Carbon and Nitrogen

DOC and DON averaged 62.5 ± 32.5 µM (n=19) and 6.1 ± 3.1 µM (n=15), respectively, during this expedition (Table S5). Note that DON was below detection limit in 4 samples. Differences were observed between the two polar regions. Near the Antarctic Peninsula, DOC exhibited a lower concentration, 57.6 ± 7.4 µM (n=7), in contrast to the Weddell Sea, with slightly higher DOC levels (77.4 ± 36.8 µM, n=12) (Table S5). Similarly, TN and DON concentrations were slightly higher in the Weddell Sea, measuring 29.1 ± 5.8 µM (n=12) and 6.3 ± 4.1 µM (n=10), respectively, compared to the Western Antarctic Peninsula, where concentrations of 27.4 ± 2.4 µM (n=7) and 5.3 ± 2.9 µM (n=5) were measured. The average contribution of dissolved amines (dMMA, dDMA, dTMA and dDEA) to DOC and DON was determined to be 0.3 ± 0.2 % (n=19) and 1.8 ± 2.8 % (n=15), respectively.

POC and PON were measured in all samples, with averages of 7.6 ± 5.3 µM (n=19) and 1.2 ± 0.9 µM (n=19), respectively (Table S5). Statistical analysis revealed significantly higher POC and PON concentrations in the Western Antarctic Peninsula (POC: 10.7 ± 7.3 µM, PON: 1.8 ± 1.2 µM, n = 7) than in the Weddell Sea (POC: 5.7 ± 1.7 µM, PON: 0.9 ± 0.2 µM, n = 12) (p=0.036 for POC and p=0.028 for PON) (Fig. S5). C:N ratio of POM closely approximated the canonical Redfield ratio of 6.6, with an observed mean of 6.4 ± 0.6 (n=19) (Table S5). The contribution of particulate TMA to POC and PON averaged 0.7 ± 0.3 % and 1.5 ± 0.6 % (n=18 for both), respectively.

### 3.4.2 Sulfur compounds

DMSP concentrations averaged 35.1 ± 16.6 nM considering all samples (n=19) (Table S5). A disparity in the concentration of this sulfur compound was observed between the Western region of the Antarctic Peninsula and the Weddell Sea, where concentrations averaged 44.8 ± 20.9 nM (n=7) and 29.4 ± 9.8 nM (n=12), respectively. Similarly, DMS, the breakdown product of DMSP, showed statistically significant differences between samples, with higher values at the Western Antarctic Peninsula (1.7 ± 0.4 nM, n=7) and lower values in the Weddell Sea (1.0 ± 0.4 nM, n=12; p=0.011) (Table S5 and Fig. S5).

### 3.4.3 Nutrients

Nutrient levels remained relatively stable throughout the duration of the cruise, with average concentrations of 21.0 ± 2.5, 0.2 ± 0.0 µM for Nitrate, Nitrite, and 54.9 ± 6.1 µM for Silicate, respectively (n=19) (Table S5). Contrastingly, Ammonium, Phosphate and TP showed statistically

significant differences within the two marine areas with higher values for Weddell Sea, $1.6 \pm 0.4$
µM for Ammonium, $2.3 \pm 0.2$ µM for Phosphate and $17.5 \pm 9.0$ µM for TP compared to the
Western Antarctic Peninsula area, $0.8 \pm 0.2$ (n=19; p<0.001), $1.9 \pm 0.3$ (n=19; p=0.0098) and 4.9
$\pm 1.9$ µM (n=19; p=0.0018).
**3.5 Multivariate statistical analysis of the distributions of alkylamines, microbiota, chemical**
**and environmental variables**
We investigated how seawater biogeochemistry influences amine concentrations to address the
largely unexplored role of microbiology and ecology in marine alkylamine cycles. Principal
Component Analyses were conducted to examine correlations among a suite of physical,
biogeochemical (including amine forms) variables and biomass data for microbial and viral
populations of the 18 sampled stations (sample #3 was excluded because pTMA was missing) (Fig.
4). Variables like dMMA, DON, V4 and nanoflagellate biomasses were excluded from the PCA
analyses because they were not detected in all samples, dinoflagellates and diatoms 10–20 µm
biomass, TN, TOC and TON were excluded because they overlapped with included variables.
Overall, the distribution of variable vectors within the multidimensional space of the PCA should
help understand how environmental and biological variables influence the variability of marine
alkylamines.
The first PCA, PCA (a), (Fig. 4a) provided an integrative perspective on the microbial community
structure, encompassing the biomass of total bacteria, virus, phytoplankton biomasses
(phytoplankton > 1 µm, including cryptophytes quantified by flow cytometry; and dinoflagellates
cysts, dinoflagellates and diatoms >20 µm biomass, determined by optical microscopy) and
biomass estimates for ciliates, assessed via optical microscopy. Additionally, it included physical
(SST, salinity, PAR) and biogeochemical (DMSP, DMS, Chlorophyll-a, $F_v'/F_m'$, POC, PON,
DOC, TP and nutrients) variables. The first two principal components (PC1 and PC2) accounted
for 57.4 % and 14.9 % of the total variance, respectively. In PCA (a), while abiotic factors (SST,
ammonium, phosphate), particulate organic matter (POC, PON) and total virus biomass were the
most significant contributors to PC1, pTMA, dDMA, DMSP, Nitrate and Silicate contributed
predominantly in a positive direction to the PC2 axis (Fig. 4a). The observed methylamines were
neither aligned with physical parameters, nor with phytoplankton biomass or chlorophyll-a, which
may be regulated by e.g. iron (Fe) availability (not measured in this study). However, they more
strongly covaried with nutrient concentrations, particularly silicate, and DMSP. Note that the
expedition took place during a transitional period, after the peak of the ice melt and associated
diatom blooms, alongside the initial stages of sea-ice formation.
Figure 4b further delves into a second analysis, PCA (b), focusing on specific biomass categories,
including phytoplankton 1–2 µm, phytoplankton 2–20 µm (containing cryptophytes), diatoms 20–
40 µm and >40 µm, dinoflagellates 20–40 µm and >40 µm, V1, V2 and V3 viral fractions, and
HNA and LNA bacteria, each of them characterized by optical microscopy or FCM. This detailed
analysis provides nuanced insights into the interplay between microbial community dynamics and
seawater biogeochemistry. The first and third Principal Components (PC1 and PC3, which were
the components that explained the largest variance of the amines) account for 54.6 % and 8.0 %
of the total variance, respectively. In summary, in PCA (b), pTMA and dDMA were aligned with
nanophytoplankton (2-20 µm) which included cryptophytes (*Cryptomonas* spp.) and not with the
biomass of larger phytoplankton (Fig. 4b).
Varimax rotation was applied to the factors extracted via Principal Axis Factoring to enhance
interpretability by maximizing the variance of Factor loadings, resulting in more distinct and
interpretable patterns (Jolliffe, 2002) using the same variables as those applied in the PCAs. All
key parameters, detailed in Table 1, were included in the analyses to support a robust interpretation
of the Principal Components. Five Factors were selected from the scree analysis, in sum explaining
69 % (Table 1a) and 71 % (Table 1b) of the total data variance, respectively. Table 1 presents the
loadings of the variables on the five rotated Factors, indicating the strength of correlation of each
variable and its respective factor. Loadings (positive or negative) above 0.2 (or below -0.2) were
considered significant. Finally, Pearson correlations for all pairs of variables are presented in Fig.
S6. Overall, the Factor Analysis reinforced the exploration of the combined contribution of
alkylamines and other variables to the total variance observed in the previous PCA analyses.
pTMA showed larger positive loadings in Factor 2 of Table 1(a) (along with nutrients and DMSP)
and Factor 3 of Table 1(b) (with nanophytoplankton and *Cryptomonas* spp. and slightly with the
V1 virus population). This suggests that pTMA mostly occurred in the nanophytoplankton size
fraction (<20 µm), that typically harbours most of the DMSP (Stefels et al., 2007). Also in the
pairwise correlation analysis (Fig. S6), pTMA was best positively correlated with phytoplankton
cells between 2 and 7 µm, *Cryptomonas* spp. (Mantel statistical test *r* and p-value of 0.71 and
0.007, respectively), silicate (Mantel statistical test $r$ and p-value of 0.63 and 0.01, respectively),
as well as with DMSP (Mantel statistical test $r$ and p-value of 0.51 and 0.034, respectively), PNF
10–20 µm (Mantel statistical test $r$ and p-value of 0.37 and 0.037, respectively), HNF and
particularly HNF 2–5 µm (Mantel statistical test $r$ and p-value of 0.49 and 0.03, respectively).
Conversely, it was negatively correlated with big diatoms (>40 µm) (p<0.1). Dissolved TMA
showed its largest negative and positive loadings in Factor 1 and 3 of Table 1(a), together with
chlorophyll-a and particulate organic matter, and Factor 1 and 4 of Table 1(b), where it was
essentially correlated with nanophytoplankton. Indeed, in the correlation matrix (Fig. S6) dTMA
correlated with phytoplankton cells between 7 and 15 µm (Mantel statistical test $r$ and p-value of
0.53 and 0.025, respectively), and more generally with phytoplankton cells ranging from 2 to 20
µm (Mantel statistical test $r$ and p-value of 0.45 and 0.004, respectively).
Dissolved DMA contributed significantly to Factor 2 in Table 1(a) and similarly in several Factors
in Table 1(b), concurring with pTMA, DMSP, photosynthetic cells in the 2–20 µm size range,
HNA Bacteria, and nutrients (particularly silicate). In the correlation matrix (Fig. S6), dDMA was
positively correlated with particulate TMA (Mantel statistical test $r$ and p-value of 0.60 and 0.029,
respectively), *Cryptomonas* spp. (Mantel statistical test $r$ and p-value of 0.65 and 0.043,
respectively), DMSP (Mantel statistical test $r$ and p-value of 0.61 and 0.017, respectively), silicate
(Mantel statistical test $r$ and p-value of 0.72 and 0.004, respectively), nanoflagellate abundances,
PNF (10–20 µm), HNF, and small HNF (2–5 µm) (Mantel statistical test $r$ and p-value of 0.52 and
0.02, respectively).
Dissolved DEA had several similar positive and negative loadings in Table 1(a), which was also
contributed by bacteria and general phytoplankton biomasses, and $F_v'/F_m'$. Additionally, dDEA
contributed principally to Factor 5 in Table 1(b) together with HNA Bacteria. In pairwise
correlations (Fig. S6), dDEA showed positive correlations with $F_v'/F_m'$ (also indicated by the
Mantel statistical test with $r$ and p-value, 0.24 and, 0.038, respectively) and DMS (Mantel
statistical test with $r$ and p-value, 0.45, 0.046, respectively), and with dinoflagellate cysts, small
dinoflagellates (10–20 µm) and big diatoms (>40 µm) (p<0.1).
Finally, dMMA, which was excluded from the PCA and Factor analysis as it was below detection
limit in most cases, is known to originate primarily from the bacterial degradation of N-containing
osmolytes and amino acids (Lidbury et al., 2015b; Mausz and Chen, 2019). dMMA exhibited a
significant positive correlation with DOC (Mantel statistical test $r$ and p-value of 0.49 and 0.016,
respectively) and TOC (Mantel statistical test $r$ and p-value of 0.48 and 0.02, respectively,) and
negative correlation with total and HNA bacteria biomass (Mantel statistical test $r$ and p-value of
-0.28 and 0.04, respectively), salinity (Mantel statistical test $r$ and p-value of -0.43 and 0.012,
respectively), and SST (Fig. S6).

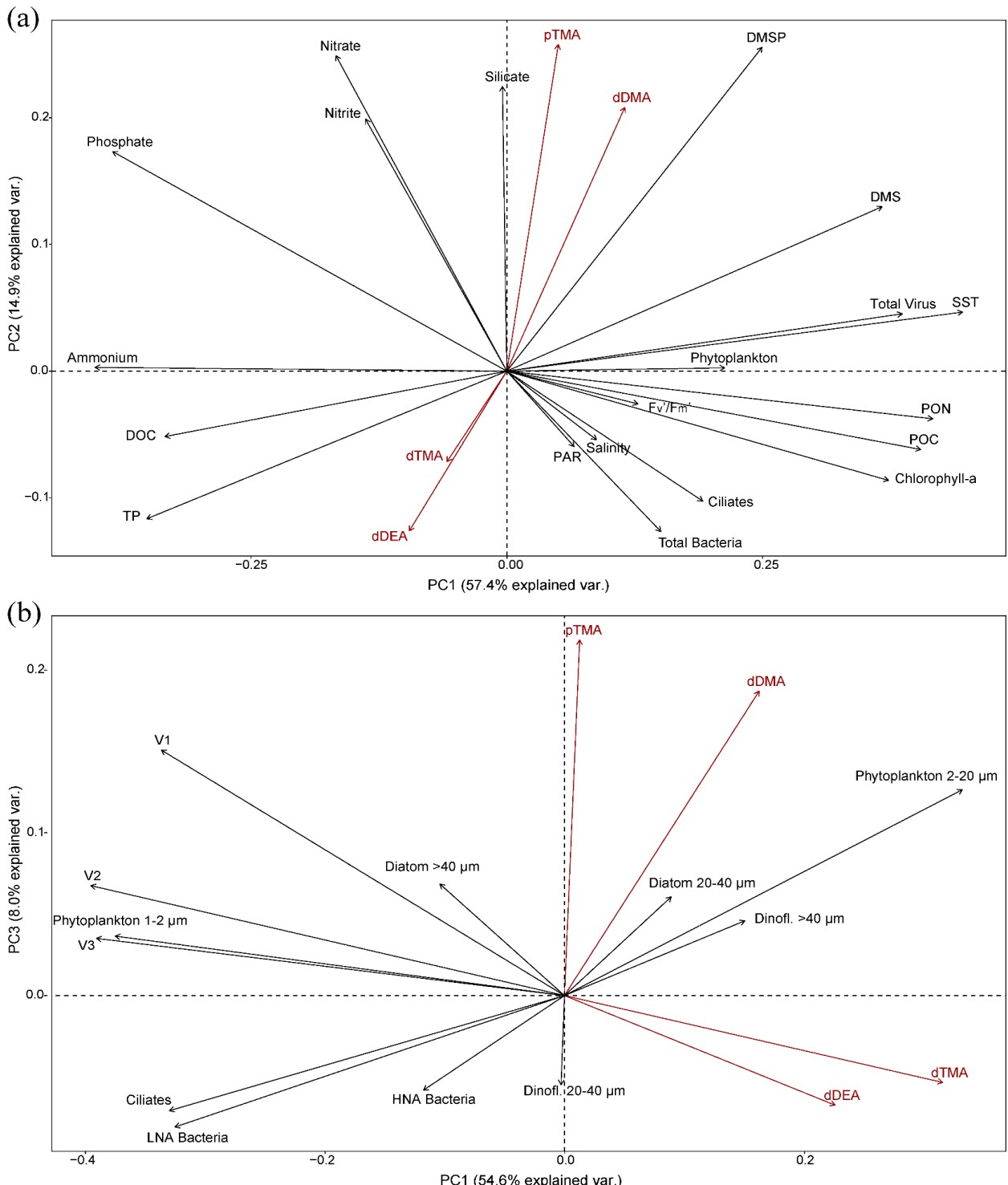


**Figure 4**. Principal component analyses of the highest explanatory biogeochemical parameters in the 18 underway
seawater samples collected (see text). (a) PC2 vs PC1 of all physical and biogeochemical data from the water samples
and the biomass of the main phytoplankton groups and viral, bacterial and ciliate biomasses and (b) PC3 vs PC1 of
the more specific PCA run considering the biomasses of size-resolved phytoplankton types and ciliates, active and
non-active bacterial cells and the virus fractions. The percentage of explained variance is given on each principal
component axis. Amine forms are in red to facilitate visualization.

**Table 1.** Factor Analysis loadings corresponding to the PCA analyses shown in Fig. 4, after Varimax
rotation. The upper part of the Table, Variables (a) refers to PCA (a) (Fig. 4a), while the bottom part refers
to PCA (b) (Fig. 4b). Loadings above 0.2 (or below -0.2) (significant loadings) are shown in italics, and
above 0.6 (or below -0.6) in bold italics. The last two lines of each table refer to the total variance explained
by one factor in the data (SS Loadings) and to the proportion of the total variance in the dataset (Proportion
Var.).

| Variables (a) | Factor 1 | Factor 2 | Factor 3 | Factor 4 | Factor 5 |
|---|---|---|---|---|---|
| pTMA | 0.1 | *0.50* | *-0.21* | 0.03 | *0.30* |
| dTMA | *-0.40* | -0.08 | *0.41* | -0.14 | 0.00 |
| dDMA | 0.11 | *0.74* | *0.44* | -0.07 | 0.09 |
| dDEA | *-0.25* | 0.01 | *0.39* | *0.32* | *-0.35* |
| Chlorophyll-a | *0.26* | -0.10 | ***0.85*** | 0.10 | *0.31* |
| SST | ***0.92*** | 0.15 | *0.24* | 0.12 | *0.29* |
| Salinity | 0.03 | 0.09 | *0.20* | ***0.96*** | -0.05 |
| $F_v$'/$F_m$' | 0.06 | 0.00 | 0.03 | ***0.72*** | 0.18 |
| PAR | -0.07 | 0.13 | *0.51* | *0.26* | -0.10 |
| DMSP | 0.10 | *0.58* | *0.25* | -0.04 | ***0.68*** |
| DMS | *0.43* | 0.16 | 0.09 | 0.05 | ***0.70*** |
| Total Bacteria | *0.47* | -0.10 | 0.12 | *0.41* | *-0.28* |
| Total Virus | ***0.87*** | 0.09 | 0.05 | 0.09 | *0.25* |
| Phytoplankton | -0.17 | -0.16 | 0.19 | *0.27* | ***0.74*** |
| Ciliates | *0.55* | *-0.28* | -0.12 | -0.05 | 0.02 |
| Nitrate | -0.13 | ***0.69*** | *-0.35* | 0.16 | 0.01 |
| Nitrite | -0.03 | *0.57* | -0.11 | *-0.24* | -0.09 |
| Ammonium | ***-0.80*** | 0.04 | *-0.25* | 0.05 | -0.15 |
| Silicate | -0.06 | ***0.91*** | *0.29* | *0.25* | 0.00 |
| Phosphate | *-0.51* | *0.55* | *-0.45* | 0.09 | *-0.22* |
| DOC | *-0.40* | 0.01 | 0.02 | *-0.48* | *-0.52* |
| PON | *0.39* | -0.01 | ***0.74*** | 0.09 | *0.42* |
| POC | *0.35* | -0.07 | ***0.74*** | 0.13 | *0.41* |
| TP | *-0.27* | -0.03 | -0.17 | 0.13 | ***-0.72*** |
| **SS Loadings** | **4.10** | **3.28** | **3.37** | **2.34** | **3.30** |
| **Proportion Var.** | **0.17** | **0.14** | **0.14** | **0.10** | **0.14** |

| Variables (b) | Factor 1 | Factor 2 | Factor 3 | Factor 4 | Factor 5 |
|---|---|---|---|---|---|
| pTMA | 0.00 | -0.14 | *0.93* | 0.02 | -0.02 |
| dTMA | *-0.30* | *0.21* | -0.18 | *-0.46* | -0.03 |
| dDMA | 0.08 | 0.09 | *0.41* | *-0.54* | *0.40* |
| dDEA | -0.19 | 0.06 | *-0.24* | *-0.4* | *0.47* |
| HNA Bacteria | 0.15 | *0.46* | -0.15 | *0.44* | *0.61* |
| LNA Bacteria | *0.29* | 0.07 | -0.05 | **0.76** | 0.15 |
| V1 | **0.85** | *0.37* | *0.30* | 0.15 | 0.04 |
| V2 | **0.87** | 0.01 | 0.10 | 0.17 | -0.06 |
| V3 | **0.81** | 0.10 | 0.04 | *0.26* | 0.02 |
| Phytoplankton 1–2 µm | **0.95** | -0.04 | -0.11 | -0.04 | -0.03 |
| Phytoplankton 2–20 µm | *-0.31* | *0.45* | *0.36* | *-0.42* | -0.05 |
| Diatoms 20–40 µm | 0.10 | *0.94* | -0.09 | -0.14 | 0.09 |
| Diatoms >40 µm | 0.18 | *0.82* | 0.03 | *0.37* | 0.09 |
| Dinofl. 20–40 µm | 0.04 | 0.04 | -0.07 | -0.05 | *-0.55* |
| Dinofl. >40 µm | 0.00 | *0.93* | -0.09 | *-0.22* | -0.05 |
| Ciliates | *0.62* | 0.00 | *-0.20* | 0.19 | -0.05 |
| **SS Loadings** | **3.84** | **3.04** | **1.51** | **1.97** | **1.15** |
| **Proportion Var.** | **0.24** | **0.19** | **0.09** | **0.12** | **0.07** |


## 4 Discussion

### 4.1 Alkylamine distributions

The almost exclusive detection of TMA in particles suggests that this may be the predominant form of methylated amines within cells. Release from cells explains that dissolved TMA is consistently present in all our seawater samples, together with the fact that TMA has the lowest Henry's constant, indicating it is the most soluble amine. This tertiary amine is known to be the primary compound released during the decomposition of marine algae and microorganisms, marsh grasses and fish, mainly as a breakdown product of quaternary amine precursors (Mausz and Chen, 2019; Sun et al., 2019). Three other alkylamines were detected dissolved in seawater. Their distributions varied across regions around the Antarctic Peninsula: the samples off the Western Antarctic Peninsula harboured different total dissolved amine concentrations (78.3 ± 44.7 nM; n=7) from those from the northern Weddell Sea (42.4 ± 24.9 nM; n=12) (Fig. 2a). This coincided

with higher Chl-a levels west of the Antarctic Peninsula (Fig. 1, Table S2 and Fig. S5). Also, $F_v'/F_m'$ values were slightly higher in samples from the Antarctic Peninsula, indicating greater photosynthetic efficiency and suggesting that in this area phytoplankton were in better physiological condition than in the Weddell Sea. Given the similarities in phytoplankton abundances and composition of the two areas, this difference can likely be attributed to the potential effect of light stress, since waters of the Weddell Sea were clearer and more stratified (data not shown), hence more exposed to excess of damaging sunlight. Most samples were collected at 18:00 local time, which corresponds to daylight hours during the Austral summer. Potential diel variations in amine concentrations should be taken into account in future studies.

Amines have been measured in seawater in polar regions primarily by Gibb and Hatton (2004), who used a flow-diffusion gas chromatography method with selective nitrogen detection in Marguerite Bay, Antarctica, and by Dall'Osto et al. (2017, 2019), with subsequent methodological improvements introduced by Akenga and Fitzsimons (2024). Gibb and Hatton (2004) reported maximum dMMA concentrations of 36 nM, while Dall'Osto et al. (2017, 2019) observed concentrations of total methylated amines (3–10 nM) that were significantly lower than those measured in the present study. Here, we found regional differences in the composition of the alkylamine mixture. In the proximity to the Western Antarctic Peninsula, dMMA was absent, with dDMA dominating, contributing up to 64 % of the total dissolved amines, followed by dTMA with a 27 % contribution and dDEA with 9 % (Fig. 2a). Conversely, samples collected within the Weddell Sea exhibited a distinct composition, with TMA comprising the highest proportion at 50 %, followed by dDMA at 27 %, dDEA at 16 % and dMMA at 7 %. In this study, dDEA concentrations are similar to the range reported by van Pinxteren et al. (2019) for the SML and seawater in tropical waters (9 to 23 nM). Overall, alkylamines can be released through various processes, including excretion by primary producers and bacterial activity, protist egestion, sloppy feeding by predators, and viral lysis (Bronk, 2002). This agrees with the fact that phytoplankton and bacteria function as DON producers (Antia et al., 1991; Bronk, 2002; Wheeler et al., 1974; Wheeler and Kirchman, 1986).

Phytoplankton release DON actively through mechanisms such as osmotic adjustments, reduced N excretion in response to changes in light, and autolysis. Phytoplanktonic passive release can occur due to physiological stress induced by factors such as ultraviolet radiation, temperature

fluctuations, and light variations, as well as interactions with microzooplankton grazing and viral
infections leading to lysis (Bronk, 2002). Viruses further contribute to DON production by
inducing host cell lysis during the final stages of infection, releasing the cellular contents into the
environment (Bronk, 2002). Similar processes are expected to occur with methylated amines (Sun
et al., 2019). Releasing N-rich dissolved organic matter (DOM) demands considerable energy from
healthy phytoplankton cells (Ward and Bronk, 2001). In the Southern Ocean, N is generally not
limiting because its use is limited by Fe and light; however, in the Western Antarctic Peninsula,
where primary production can likely be supplied with Fe and other micronutrients from land,
inorganic N may become depleted in phytoplankton blooms reaching limiting levels, as observed
in Dittrich et al. (2022). Under these specific conditions, phytoplankton can also act as DON
consumers, and the recycling of phytoplankton-released DON may provide an essential,
bioavailable N source for sustaining phytoplankton growth. Notably, it has been reported that
phytoplankton like the chlorophyte *Platymonas* (phototrophic nanoflagellate) incorporate primary
amines from natural seawater efficiently, potentially supporting robust growth (North, 1975).
Similarly, diatoms have demonstrated efficient uptake of alkylamines (Wheeler and Hellebust,

695    1981).

Bacteria are identified as the primary consumers and transformers of organic matter, as evidenced
by the relationships between bacterial abundance and DON and DOC concentrations (Fig. S6).
Furthermore, methylamine-degrading bacteria play a crucial role in releasing bioavailable N from
alkylamines, which supports diatom growth, while diatoms provide organic C to bacteria in a
mutualistic exchange (Stein, 2017; Suleiman et al., 2016). Moreover, marine bacteria metabolize
methylamines as a N source via different pathways facilitating direct assimilation of N into
biomass (Lidbury et al., 2015b; Sun et al., 2019; Taubert et al., 2017). This recycling of amines
may explain their nanomolar concentrations in seawater, suggesting they may serve as valuable
organic N sources for both phytoplankton and bacteria. Given their volatile nature, alkylamines
are also expected to be lost to the atmosphere. The cycle of methylated amines shares several
similarities with the cycles of methylated sulfur compounds, such as DMSP and DMS, in the
marine environment. Both methylated amines and sulfur compounds originate from marine
phytoplankton and participate in atmospheric processes. Recent studies have shown that TMA
monooxygenase, an enzyme in marine bacteria, can oxidise both TMA and dimethylsulfide (Chen
et al., 2011; Lidbury et al., 2016). Thus, parallelisms between marine methylated amines and DMS
metabolism underscores the importance of studying these molecules in tandem.

**4.2 Correlations between alkylamines, chemical and environmental variables, and the microbial community**

Using PCA, Factor Analysis, pairwise correlation analyses and statistical Mantel test, we found
that TMA appears to be predominantly produced intracellularly by nanophytoplankton.
Subsequently, it is released into the environment through cellular stress, mortality, or even by
mechanical processes like filtration during sampling. This could explain the observed pairwise
opposite correlation between particulate and dissolved TMA. The production of TMA is likely
linked to the enzymatic activity of TMAO reductase (Mausz and Chen, 2019), an enzyme which,
like dimethyl sulfoxide reductase (Spiese et al., 2009), occurs in marine bacteria but is potentially
common in phytoplankton cells too. This enzyme reduces TMAO, a prevalent osmolyte like
glycine betaine in phytoplankton (Gibb and Hatton, 2004). Dissolved DMA appears to exhibit a
causal relationship with particulate TMA, suggesting a shared phenomenology or a common
origin. The statistical associations suggest that dDMA is linked to nanophytoplankton, potentially
originating from the degradation of TMA or TMAO by bacteria or phytoplankton themselves. In
aerobic conditions, DMA is produced from TMAO via TMAO demethylase (Barrett and Kwan,
1985; Lidbury et al., 2014). Although TMAO demethylase activity has not yet been reported in
phytoplankton, its presence in fish tissues (Kimura et al., 2000) and the direct evidence of TMAO
occurrence in polar diatoms (Dawson et al., 2020; Fitzsimons et al., 2024) suggest that this enzyme
may also occur in eukaryotic microalgae. Although the involvement of a TMAO demethylation or
any other enzyme requires genomic confirmation, our findings suggest that phytoplankton could
directly release DMA or indirectly through bacteria attached to the outer membrane or residing in
the phycosphere. In tropical waters, van Pinxteren et al. (2019) reported positive correlation
between amines and pigments (fucoxanthin and chlorophyll-a) suggesting that amine production
was fuelled by algal metabolism, most likely diatoms. In our study in polar waters, we found that
TMA and dissolved DMA were closely related to nanosized phytoplankton.
Overall, dDEA exhibited an inverse correlation with particulate TMA. Notably, dDEA did not
display a strong distributional alignment with any specific microbial variables, although a weak
association with active bacteria was observed. Additionally, dDEA showed a moderate positive

740 correlation with the photosynthetic efficiency of phytoplankton cells ($F_v'/F_m'$) and with different

741 phytoplankton groups compared to MAs. As expected, $F_v'/F_m'$ displayed an inverse relationship

742 to nutrient availability. In the Southern Ocean, $F_v'/F_m'$ declines when iron availability limits

743 primary productivity despite the presence of elevated macronutrient concentrations (Wu et al.,

744 2019). Although the precise source of dDEA remains unclear, these findings demonstrate that DEA

745 is widespread in Antarctic waters and follows distinct biological and biogeochemical pathways

746 compared to MAs. We speculate that DEA may be formed by degradation of an amino acid

747 precursor, potentially proline, considered an important N-bearing osmolyte (Fitzsimons et al.,

748 2024). However, further research is needed to identify its specific origins and the processes

749 governing its distribution. Finally, regarding dMMA, the labile and volatile nature of this

750 compound suggest that bacteria efficiently remineralize dMMA into ammonium (Lidbury et al.,

751 2015b), and that MMA volatilizes quickly to the atmosphere, both processes contributing to the

752 rapid depletion of MMA in surface waters. Zhang et al. (2023) demonstrated that elevated salinity

753 enhances the tendency of amines to volatilize from surface seawater by suppressing amine

754 ionisation, thereby increasing exchange fluxes.

755 Altogether, the multivariate and pairwise correlation analyses make us concur with previous works

756 in that phytoplankton are the primary producers of amines or amine precursors (Fitzsimons et al.,

757 2023; van Pinxteren et al., 2019; Poste et al., 2014). However, we identify nanophytoplankton and

758 *Cryptomonas* spp. populations, instead of diatoms, as the main responsible for TMA and DMA

759 production in Antarctic waters in late summer. Smaller phytoplankton, likely those that are better

760 adapted to thrive under iron-limited conditions (Schoffman et al., 2016), would synthesise and

761 harbour most of the intracellular TMA. Part of it would be released likely through processes such

762 as cell mortality or through physiologically-driven DOM excretion. Likewise, DMA was

763 statistically associated with small phytoplankton cells and heterotrophic nanoflagellates (PNF and

764 HNF, respectively) as well as DMSP, exhibiting a distribution similar to the sulfur osmolyte. DMA

765 was more closely associated with phytoplankton than with bacteria, which are expected to be

766 responsible for TMA demethylation into DMA. This suggests that DMA is largely produced from

767 phytoplankton TMA or TMAO by the algal cells themselves or closely associated bacteria. Finally,

768 the distribution of DEA suggests distinct biogeochemical pathways compared to methylamines,

769 potentially involving larger phytoplankton and bacterial communities. Notably, the factor most

770 strongly linked to mortality, viruses, did not appear to influence alkylamine pathways. However,

incorporating viral lysis as a key phenomenon in Antarctic phytoplankton dynamics is essential
for advancing the understanding of microbial interactions and improving the accuracy of organic
matter flux estimations in this climate-sensitive region (Biggs et al., 2021).
Our findings indicate that alkylamine distributions are linked to microplankton trophic webs, in
particular to certain phytoplankton cell size groups and ecophysiological conditions rather than to
total biomass. Our approach does not allow us to quantify how much of the amines are produced
directly by phytoplankton or through bacterial reworking of phytoplankton metabolites, yet we
provide indications that both processes occur. Dissolved and particulate alkylamines accounted for
non-negligible proportions of DON (ca. 1.8 %, with a maximum of 8.7 %), and of PON (ca. 1.5
%, with a maximum of 3.1 %). These proportions are reported here for the first time, providing a
novel insight into the quantitative contribution of alkylamines to marine organic N pools.
This study contributes to the necessity of increasing alkylamine determinations to be incorporated
into future biogeochemical and climate models, given the pivotal role of alkylamines in both
marine and atmospheric systems. In the Southern Ocean, biogenic emissions influence aerosol
numbers through primary and secondary pathways, potentially enhancing CCN concentrations and
modulating cloud albedo, thereby impacting regional radiative forcing (McCoy et al., 2015). Low-
molecular-weight alkylamines contribute to both new particle formation (Brean et al., 2021) and
aerosol growth, particularly in air masses passing over melting sea ice (Dall'Osto et al., 2017).
Incorporating alkylamines in climate models for this climate-sensitive region requires gaining
understanding of their distribution and drivers. The present study represents a step forward towards
this aim.
**5 Conclusion**
Alkylamines are seawater compounds whose role as precious organic nutrients in N transfer among
trophic levels is starting to emerge. Despite their increasingly recognized importance, the
distribution, biological sources, formation mechanisms, and emission strength of marine amines
remain poorly known. This study provides several significant advances in the knowledge of the
drivers of marine alkylamine concentrations and speciation. Overall, our results emphasise that
alkylamines are embedded within marine microbial food webs, where phytoplankton, bacteria and
viruses are interconnected, thereby influencing nutrient cycling, microbial dynamics, and the
overall health of marine ecosystems. Our study, conducted under varying biogeochemical
conditions, reveals that tri- and dimethylamine present in Antarctic surface waters were primarily
sourced from nanophytoplankton cells and the associated bacteria and heterotrophic
nanoflagellates, and diethylamine from hitherto unknown processes. Describing the distribution
and behavior of alkylamines in the surface ocean is pivotal for understanding their roles in marine
biogeochemical cycles, atmospheric chemistry, and climate.

## 6 Author Contributions

AR, MD'O, RS, and EB conceptualized and designed the study. AR and AS collected seawater
and amine samples during the PolarChange Expedition. AR, under the supervision of MFF and
PA, processed and analyzed the amine samples, generating the amine dataset. MFF provided
essential resources for the amine analysis. ELS, QG, MV, DV, CW, RS, and EB participated in
the expedition, collected samples, and conducted biogeochemical and biological analyses. YMC
and AR processed and analyzed flow cytometry samples at ICM. AR performed the statistical
analyses, prepared the figures, and drafted the manuscript's first version. AR, MFF, PA, CW, RS,
and EB contributed to data interpretation and manuscript writing. All authors reviewed, revised,
and approved the final version of the manuscript.

## 7 Data availability

All data are provided in the Supplementary Information file.

## 8 Competing Interests

The authors declare that they have no conflict of interest.

## 9 Acknowledgements

We would like to thank the crew of the RV *Hesperides* for the logistic support, making possible
the data collection of this study. Special thanks to Mara Abad and Núria González Fernández for
TOC, TN and nutrient analyses at the Chemistry Service of the ICM-CSIC. We thank Jair Antonio
Arévalo Lirio and Sofía Ibáñez Homedes for assistance counting flagellates and bacteria.

**10 Financial support**
AR was supported by the FPI grant (PRE2020-092994) from the Spanish Ministerio de Ciencia e
Innovación (MICIN) and European Social Fund (ESF) 'Investing in your Future'. The
POLARCHANGE project (PID2019-110288RB-I00) also received funding from the Spanish
Ministerio de Ciencia e Innovación (MICIN). Further support was provided through an Advanced
Grant from the European Research Council (ERC-2018-AdG #834162). This study is part of the
POLARCSIC platform activities, and had the institutional support of the 'Severo Ochoa Centre of
Excellence' accreditation (CEX2019-000928-S) to the ICM-CSIC.

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
