# Peer review of "Distribution of alkylamines in surface waters around the Antarctic"

_EGUsphere, 2025_

## Author Comment (AC1)

**REVIEWER 1**: This study presents a comprehensive investigation into the distribution, sources, and biogeochemical drivers of alkylamines in Antarctic surface waters, offering novel insights into polar marine nitrogen cycling and microbial ecology. The authors employ a high-sensitivity analytical method (HS-SPME-GC-NPD) to quantify dissolved and particulate alkylamines, addressing long-standing challenges in measuring these reactive, low-concentration compounds. These findings are timely, given the climate relevance of alkylamines in aerosol formation and their underexplored roles in polar microbial food webs.

Authors: We thank the reviewer for his/her positive comments and appreciations.

Key Concerns:

1. In the introduction, lines 57-60, the authors have elaborated extensively on aerosol-related content and cited relevant references. However, no discussion on aerosol is present in the discussion section. It is recommended to include a discussion on aerosol in this section. Incorporating wind speed/SST data could further quantify their climate relevance.

Authors: We agree with the reviewer and we have added the following sentences at the end of the discussion "These findings also highlight the necessity of increasing alkylamine determinations to be incorporated into future biogeochemical and climate models, given the Southern Ocean's abundance of aerosol precursor gases and the pivotal role of alkylamines in both marine and atmospheric systems. Biogenic emissions influence atmospheric chemistry through primary and secondary pathways, potentially enhancing CCN concentrations and modulating cloud albedo, thereby impacting regional radiative forcing (McCoy et al., 2015). The contribution of low-molecular-weight alkylamines to aerosol mass, particularly from air masses passing over melting sea ice (Dall'Osto et al., 2017), underscores the critical role of marine plankton and sea-ice melt in particle formation and Antarctic climate regulation (Brean et al., 2021). Our results emphasize the need for an understanding of ocean–atmosphere interactions in pristine polar environments and, more broadly, of aerosol processes that likely play a major role in climate dynamics.". We are unable to calculate amine fluxes because, although we have sea surface temperature and wind speed data, we lack the concentrations of amines in the air.

McCoy et al., 2015: 10.1126/sciadv.1500157. The other references are already cited in the text.

2. The hypothesis that phytoplankton directly produce DMA via TMAO demethylase (Lines 691–693) lacks direct evidence. While the authors cite fish tissue studies (Kimura et al., 2000), extrapolating this to microalgae requires genomic or enzymatic validation.

Authors: Thanks for allowing us to clarify. TMAO is a cellular osmolyte and is present throughout marine ecosystems, from surface seawaters to deep sediments (Fitzsimons et al., 1997; Gibb et al., 1999; Carpenter et al., 2012, Mausz et al., 2019) and also in polar diatoms (Dawson et al., 2020, Fitzsimons et al., 2024). The fact that eukaryotic cells like those of some fish have the capability to enzymatically transform TMAO to DMA allows us to speculate that this may also occur in phytoplankton. We know this is just speculation and we are trying to provide an explanation for

the relationship between DMA and phytoplankton. We have modified the sentence: "Although the specific enzymatic pathways are unknown, it is plausible that phytoplankton could directly release DMA or indirectly through bacteria attached to the outer membrane or residing in the phycosphere."

Dawson et al., 2020: 10.1093/icb/icaa133; Fitzsimons et al., 1997: https://doi.org/10.1016/S0146-6380(97)00062-4; Gibb et al., 1999: https://doi.org/10.1016/S0967-0645(98)00119-2; Carpenter et al., 2012: https://doi.org/10.1039/c2cs35121h.

3. Despite measuring viral abundances (V1–V3), the discussion overlooks viral lysis as a potential amine source. Given the correlation between Factor 1 and V1–V3 (Table 1b), a brief analysis of virus-phytoplankton interactions (e.g., *Phaeocystis* lysis) would strengthen the narrative.

Authors: We agree with the reviewer; however, in lines 743-744 we had written "Notably, the factor most strongly linked to mortality, viruses, did not appear to influence alkylamine pathways." Factor Analysis enabled us to numerically visualize the correlations between variables. Although Factor 1 is associated with V1–V3, no amines appeared to be linked to the virus populations. From the Factor Analysis, the only correlation between viruses and amines has already been stated in lines 652–654. We have added this sentence "However, incorporating viral lysis as a key phenomenon in Antarctic phytoplankton dynamics is essential for advancing the understanding of microbial interactions and improving the accuracy of organic matter flux estimations in this climate-sensitive region (Biggs et al., 2021)."

Biggs et al., 2021: https://doi.org/10.1038/s41396-021-01033-6.

4. The speculated role of iron in regulating alkylamine distributions (Lines 588–589, 644–646) lacks supporting Fe concentration data.

Authors: We are trying to discuss possible explanations for the differences in amine concentrations in the Antarctic Peninsula versus the Weddell Sea. However, we agree with the reviewer that point to iron when there is no measurement may be going too far, and have made changes in the revised manuscript: "Given the similarities in phytoplankton abundances and composition of the two areas, this difference can likely be attributed to the potential effect of light stress, since waters of the Weddell Sea were clearer and more stratified (data not shown), hence more exposed to excess of damaging sunlight." The other sentence has been deleted.

5. Contrasting Antarctic alkylamine levels with prior Arctic or temperate studies would highlight polar-specific processes.

Authors: Thanks for the suggestion. Indeed, reviews by Poste et al., 2014 (including freshwater studies) and more recently by Fitzsimons et al. (2023 and 2024) report alkylamine data. Amines have been detected in various marine environments such as the Arabian Sea (Gibb et al., 1999a, b), the North Atlantic, and Cape Verde waters (van Pinxteren et al., 2012, 2019), as well as in the

ice-associated waters of the Southern Ocean (Gibb et al., 2004, Dall'Osto et al., 2017, 2019). Notably, there is currently a lack of studies on alkylamine detection in (polar) seawater, with more extensive research on alkylamines conducted in the (polar) atmosphere (Köllner et al., 2017, Brean et al., 2021). We have added these sentences: "Amines have been measured in seawater in polar regions primarily by Gibb et al. (2004), who used a flow-diffusion gas chromatography method with selective nitrogen detection in Marguerite Bay, Antarctica, and by Dall'Osto et al. (2017, 2019), with subsequent methodological improvements introduced by Akenga and Preston (2024). Gibb et al. (2004) reported maximum dMMA concentrations of 36 nM, while Dall'Osto et al. (2017, 2019) observed concentrations of total methylated amines (3–10 nM) that were significantly lower than those measured in the present study."

Gibb et al., 2004: https://doi.org/10.1016/j.marchem.2004.04.005; Gibb et al., 1999a,b: https://doi.org/10.1016/S0967-0645(98)00119-2 and https://doi.org/10.1029/98GB00743; van Pinxteren et al., 2012: dx.doi.org/10.1021/es204492b.

> 6. The particulate amine protocol (Lines 166–172) lacks critical details: (i) filter storage duration before analysis, (ii) CPA spike recovery rates, (iii) NaOH volume for amine liberation. Clarify to ensure reproducibility.

Authors: We agree with the reviewer. We have improved the section accordingly: "We also measured amines in particulates retained on GF/F filters after seawater filtration (section 2.2). Analyses were conducted ~6 months after sample collection. Prior to extraction, each filter was placed in a 20 mL autosampler glass vial and allowed to thaw inside the vial (one filter per vial). Subsequently, we added 250 µL of CPA (20 nM final concentration) as internal standard and 500 µL of 10 M NaOH, to liberate gaseous amines, and the vial was tightly sealed. This treatment was assumed to volatilize the target analytes into the vial headspace in a manner analogous to dissolved samples. Particulate amine concentrations were quantified using standard amine solutions, as described previously.". Recovery data for CPA in particulate samples are not available. We assume that the volatilisation of amines from the NaOH-treated filter is the same as that from NaOH-treated solution. Particulate amine concentrations were calculated using the same standards as for dissolved amines. This procedure was most suitable and applicable for particulate quantification. Note that this is the first time that particulate amines are analyzed in seawater.

7. Inconsistent use of "nanophytoplankton" and "phytoplankton" (2–20 µm in Methods vs. 2–7 µm in Results). Standardize definitions or justify size-class divergences.

Authors: Nanophytoplankton encompasses all phytoplankton within the size range of 2 to 20 µm. Therefore, referring to phytoplankton in the range of 2 to 7 µm still classifies them as nanophytoplankton although we wrote phytoplankton "cells 2–7 µm in size". When we consider the nanophytoplankton from 2 to 20 µm, we refer to the sum of the abundances or biomasses from 2 to 20 µm (sum of biomass or abundances between 2–7 µm, 7–15 µm and 15–20 µm). We have revised the sentences throughout the text for improved clarity.

In summary, the paper by Rocchi et al. (2025) presents valuable data and insights into the distribution and sources of alkylamines in Antarctic surface waters. With some revisions to address the issues outlined above, the paper has the potential to make a significant contribution to the field of marine biogeochemistry.

---

## Author Comment (AC2)

**REVIEWER 2**: **General comments**

This valuable manuscript describes the concentrations and distribution of four alkylamines in seawater (dissolved MMA, DMA, TMA, and DEA) and TMA in particulates. I really enjoyed reading it and I hope my comments help improve it for prompt publication. The paper summarizes extensive characterization work in a key location to understand ecosystem dynamics in polar environments and their implications for atmospheric processes. As the authors state, this is the first time TMA is measured in Antarctic waters, and the proportion of these alkylamines in DON and PON is estimated. An explanation for the presence of TMA and DMA is discussed.

Authors: We thank the reviewer for the constructive comments that helped to improve substantially the quality of the manuscript.

After carefully reviewing the manuscript, I have three general comments that support my review:

1. Methodological limitations need to be clearly stated. One important consideration that authors should add to their methods and discussion is what measures were taken (or not) to minimize the loss of alkylamines during sampling and processing.

Authors: We minimized headspace in the collection-falcon tubes by filtering until the sample volume was fully filtered/processed before tightly sealing the vials. A better description is provided now: "Seawater was collected into 50 mL propylene tubes (Falcon type), which were completely filled. For dissolved amine analysis, seawater was filtered through a 47 mm GF/F filter (0.7 µm pore size) by gravity (ca. 60 minutes, filtration timing depended on the microbial biomass and particulate matter contained in the sampled water) and directly collected into a new 50 mL propylene tube until completely filled. This procedure minimised headspace as indicated by Akenga and Fitzsimons (2024). This filtered water was preserved with concentrated 37 % HCl (analytical grade) at 1 % (v/v) final concentration. The tube was tightly closed and stored in the dark at 4 °C until analysis. In turn, after filtration, the GF/F filter was stored in a 2 mL eppendorf tube at -80 °C for particulate amine analysis.".

Have you considered losses in the underway system? Pumping and pressure changes may create turbulent flow and bubbles that negatively impact the water concentrations of these volatile compounds despite their solubility.

Authors: To assess potential losses, we looked at other VOC compounds that are more volatile than amines and observed no significant differences between the concentrations measured from the underway system and those from the surfacemost bottle (4 m) of simultaneous CTD coasts (Wohl et al., manuscript in preparation).

I understand this type of difficult measurement has limitations and they are extremely valuable in spite of them, but some acknowledgment of these pitfalls is required. Except for DEA, boiling points for MMA, DMA, and TMA are below 298K (267, 281, and 275 K respectively). More importantly, this could negatively impact your filter concentration if

they were left to dry for a long time (this period is not stated anywhere in the text). Have method accuracy and precision been determined for this method in past work? If so, state it in line 141. If not, some recovery data is needed from known standards, especially for the determination of particulate alkylamines.

Authors: This study represents the first instance of measuring particulate alkylamines; however, dissolved amines have been previously analyzed by van Pinxteren et al. (2019), Dall'Osto et al., (2019). The method that Dall'Osto et al. (2019) used was improved by Akenga and Fitzsimons (2024) and is the one used in this study. In the first version of the submitted manuscript, we erroneously wrote that the filters were left to dry; in fact, after gravity filtration the filters were stored in eppendorf tubes at -80°C. We have indicated it correctly in the revised version. Regarding precision, each dissolved amine sample was analyzed in methodological duplicates or triplicates. For replicates the median of the standard deviations is 0.07nM for dMMA, 3.4nM for dDMA, 2.3nM for dTMA and 0.5nM for dDEA. Accuracy remains challenging to determine due to multiple influencing factors, including solubility, volatility, and polarity. Recovery data for CPA in particulate samples are not available. We assume that the volatilisation of amines from the NaOH-treated filter is the same as that from NaOH-treated solution. We ensured consistency by using the same final CPA concentration as an internal standard, matching the concentrations of the working standards.

2. Figures could be improved substantially. I strongly recommend using a different type of chart, e.g. bar chart, for Figures 2 and 3. Pie charts are misleading, small concentrations are overrepresented in samples where the total concentration is low, and large concentrations are underrepresented in samples where the total concentration is high. E.g. in lines 457-459 concentrations are close but from Figure 2 it looks like dino cysts (red) are much higher in the Weddell Sea. Moreover, Figure 3 states that HNF and PNF were lost for four samples and still these measurements are included in the grand totals for the rest of the samples. This is not an acceptable approach if you are using pie charts. Figure 4, Table 1, and Figure 5 show virtually the same information. I don't think all three are needed in the main text, especially Table 1 and Figure 5 which are not reader-friendly or impactful enough in my opinion.

Authors: We thank the reviewer for these suggestions. In the text, we refer to cell abundances of each particular taxa / group, whereas the figure displays only their corresponding biomasses (as Carbon). Regarding Figure 2, pie charts provide a clear picture and therefore we recommend maintaining the current format. However, we agree with the reviewer's suggestion for Figure 3 and it is now presented as bar plots for protist biomass. Asterisk (*) in black and orange indicate where nanoflagellate data are unavailable, i.e. they should not be interpreted as zero concentrations.

[Figure]

While Figures 4 and Table 1 present similar information, both are frequently referenced throughout the results and discussion sections. Given their relevance, we consider them essential and recommend keeping them in the manuscript. To facilitate the visualization of amines, their corresponding arrows are coloured red (Figure 4). Additionally, as suggested, we have moved Figure 5 to the Supplementary Information.

[Figure]

3. The manuscript provides an extensive quantitative description of results to the detriment of a richer discussion section. A lot of comparisons and correlations are shown but few are explored in depth or with appropriate discussion of their actual significance. The scope of

Biogeosciences embraces the study of interactions between biological, chemical, and physical processes. The first paragraph of the introduction states the relevance of these compounds for atmospheric processes, which in my opinion is one of the main reasons why the scientific community is interested in measuring their concentration, but then this is not further discussed after presenting the results. There is no mention of big-picture implications so the relevance of the paper's findings, which are clearly stated in the abstract but not in the main text, is lost in the weeds. The manuscript could benefit from some discussion in the context of key references such as McCoy et al., (2015) https://www.science.org/doi/10.1126/sciadv.1500157 (just to give an example, no need to include this one in particular). One suggestion: organize the Discussion subsections according to your main findings instead of continuing with the descriptive tone that belongs to the Results.

Authors: We agree with the reviewer and we have expanded the discussion to better highlight the broader implications of alkylamines in the marine and atmospheric systems. Specifically, we have integrated references to key studies on aerosol-cloud interactions and biogenic emissions, emphasizing the role of alkylamines in CCN formation and potential impacts on cloud albedo and radiative forcing. The following sentences have been added at the end of the discussion: "These findings also highlight the necessity of increasing alkylamines estimations to be incorporated into future biogeochemical and climate models, given the Southern Ocean's abundance of aerosol precursor gases and the pivotal role of alkylamines in both marine and atmospheric systems. Biogenic emissions influence atmospheric chemistry through primary and secondary pathways, potentially enhancing CCN concentrations and modulating cloud albedo, thereby impacting regional radiative forcing (McCoy et al., 2015). The contribution of low-molecular-weight alkylamines to aerosol mass, particularly from air masses passing over melting sea ice (Dall'Osto et al., 2017), underscores the critical role of marine plankton and sea-ice melt in particle formation and Antarctic climate regulation (Brean et al., 2021). Our results emphasize the need for an understanding of ocean–atmosphere interactions in pristine polar environments and, more broadly, of aerosol processes that likely play a major role in climate dynamics." Furthermore, we have restructured the discussion and the results as suggested by the reviewer.

**Specific comments**

Introduction:

1. There is no background on why diethylamine (DEA) is measured in this study. Why is it relevant?

Authors: Diethylamine (DEA) has been analyzed in seawater (Poste et al., 2014; Van Pinxteren et al., 2012, 2019, Fitzsimons et al., 2024); however, it has been more extensively studied in aerosols (Facchini et al., 2008; Dall'Osto et al., 2019). Despite its detection in both marine water and the atmosphere, there is currently no direct evidence identifying its biological or chemical sources within the marine environment. We have added this sentence in the introduction: "Concerning the

secondary amine, DEA, the information on production pathways, potential biological precursors, or transformation processes in seawater is limited.". There is a lack of studies on DEA production pathways, potential biological precursors, or transformation processes in seawater. Given that DEA and other alkylamines and methylamines can be transferred from the ocean to the atmosphere, investigating their marine sources and cycling is crucial for assessing their role in atmospheric chemistry and cloud formation.

Methods:

1. The date of the expedition is unclear. Line 116 says 2024 but Figure 1 says 2023. Also, state in Figure 1 what was the period considered for the plot. Are the average concentrations over that time period represented?

Authors: We are sorry about this mistake. We have corrected accordingly. 2023 is the correct year. Figure 1 presents the ranges of sea surface temperature and chlorophyll concentrations observed during March 2023, coinciding with the period of our expedition. We have decided to include March (and not February) because more samples were collected in March.

2. Why were samples taken at 18:00 local time? Have you considered the effect of diel cycles (if any) in your measurements? I understand that the reason might be a logistic issue but I think some consideration is needed.

Authors: We agree with the reviewer; however, our primary focus was on investigating the sources of these compounds. While diel variability is an important aspect, similar to the diel cycles observed for DMSP and DMS, it remains an area for future research. Notably, during the austral summer, 18:00 local time still falls within daylight hours. Our sampling time was primarily determined by logistical needs, as multiple measurements were conducted simultaneously, requiring a coordinated and efficient approach. We have added a sentence "Most samples were collected at 18:00 local time, which corresponds to daylight hours during the Austral summer. Potential diel variations in amine concentrations should be taken into account in future studies.".

3. Where was the radiometer located? At the surface or near the underway system inlet?

Authors: We used the radiometer model PRR-800 to measure solar radiation at the surface level. It was located in the upper deck. We corrected the radiometer model in the text.

4. Explain how the limit of detection was calculated in line 184.

Authors: We used the lowest detectable concentration.

5. Figures S1-S5 should be updated in a better quality. The text is too small to read in most cases. In Figures S3 and S4, the axes don't match the text (lines 213 and 222 "side scatter (SCC) versus green fluorescence"). Caption of figures should clearly explain acronyms and other data in the Figures. What do percentages mean in Figure S2?

Authors: We have corrected the text and pictures and improved their quality. In the revised document, we have also corrected and added the explanations of the acronyms in the caption of the respective figures. The percentages of Figure S2 represent the population relative to the total particle count, including bacteria and all background noise; however, we have eliminated them from the corrected figure. We have also explained the acronyms in the revised manuscript like this: "SSC: Side scatter, type of light dispersion; FL1: SYBRGreenI fluorochrome, FITC/1, fluorescence captured with 536/40nm filter; FL2: Phycoerythrin natural fluorochrome, PE/1, fluorescence captured with 590/50nm filter, and FL3: Chlorophyll-a natural fluorochrome, PE-Cy5/1, fluorescence captured with 675/20nm filter."

[Figure]

[Figure]

[Figure]

[Figure]

6. Line 277: was the volume taken into account for POC and PON calculations? Please state so if variable volumes were sampled.

Authors: The volume of filtered sampled seawater for POC and PON analysis varied depending on the biomass of the sample (already written in the text, line 277). Accordingly, we accounted for the specific volume of seawater filtered for each sample to calculate POC and PON concentrations.

7. In sections 2.4.3 and 2.4.4, authors recurrently used the word "estimated". Are these measurements or estimations? Why? Please clarify.

Authors: We used the term "estimation" as a synonym of "determination" and "quantification" for most biological parameters for which no standards are available (e.g. cell counts) or when the "measured" parameters are a proxy of the targeted parameter (or variable). However, following the reviewer's question we have changed "estimated" to "determined" or "quantified" when referring to the concentration of several parameters in the new version of the manuscript.

Results:

1. Line 348: the range is incorrect compared to Table S1.

Authors: We have corrected Table S1.

2. Line 349-350: state in the text the values are the average +/- standard deviation.

Authors: We have stated it in the text as suggested.

3. Please avoid the nuances in comparisons when no statistical/quantitative measure is provided. Keep in mind that some of these comparisons you make are hard to grasp when there is no visual representation since only significantly different pairs are included in Fig. S5 (and I agree with your choice). E.g. Line 376: how do you prove the "more even distribution"? The word "slightly" is used a lot throughout the text, sometimes to describe allegedly small differences and sometimes differences of almost 100% (e.g. line 398). Please rephrase. Same with "small" in line 507.

Authors: We now refer to the coefficient of variation to prove the "more even distribution": "In this study, dDEA had the most even distribution of all alkylamines (excluding dMMA), with a coefficient of variation of 23%, compared to 101% for dDMA and 73% for dTMA.". Additionally, we have corrected and improved the text as suggested by the reviewer.

4. Line 397: I'm confused because the text states that V4 was only present in sample 15. However, Figure S2 shows the location of V4 in Sample 14.

Authors: Sorry for the mistake. It is sample 15 also in Figure S2. It has been corrected.

5. Line 465-467: Please revise. "In contrast"? diatoms 10-20 µm are larger in the Antarctic Peninsula according to Figure S5. I think something may be off with your values.

Authors: We have changed "In contrast", which was added erroneously, to "Similarly".

6. Line 527-529: state somewhere the list of variables that you left out of the PCA. You mention DON and nanoflagellate but I also count at least dMMA.

Authors: We have added this sentence to be clearer: "Variables like dMMA, DON, V4 and nanoflagellate biomasses, TN, TOC and TON were excluded from the PCA analyses.".

7. Line 543: why did you choose to plot PC1 and PC3 (Fig 4b) when you could explain more variance with PC2? Please explain this.

Authors: The focus of the article is on amines, and in PC2, amines showed lower variance. For this reason, we chose to plot PC3, instead of PC2, in which alkylamines are much more represented.

8. Maybe I am missing something but I don't quite understand why the percentages assigned to PC1 and PC2 (and PC1 and PC3 in b) do not match the Proportion Var. row in Table 1. Are they not representing the same thing?

Authors: Both analyses (PCA and Factor Analysis) represent total variance, but there is a key difference in how variance is treated in each analysis. In Factor Analysis (Table 1), the extracted Factors explain the shared variance within the dataset. Unlike PCA, these Factors are not necessarily orthogonal. However, Varimax rotation is applied to enhance interpretability by maximizing the variance of Factor loadings across variables. This rotation helps clarify the association between Factors and specific variables, which may lead to differences in the variance percentages compared to PCA.

Discussion:

1. Line 573-574: Unfortunately, you cannot state this confidently without further investigation of the losses of other alkylamines or confirmation on the sources. Coincidently, TMA has the lowest Henry's constant (9.5 mol/kg bar) which makes it less soluble in water. I anticipate TMA would also be more likely to partition to the particle phase. Please rephrase.

Authors: TMA has the lowest Henry's constant, which supports our statement that it is highly soluble and, probably for this reason, it is present in all the samples. TMA is the least volatile amine. Additionally, Henry's constant does not tell about partitioning to the particulate phase.

2. Line 586: without iron measurements or more literature data this is not a statement that you should make.

Authors: We have changed the sentence accordingly: "Given the similarities in phytoplankton abundances and composition of the two areas, this difference can likely be attributed to the potential effect of light stress, since waters of the Weddell Sea were clearer and more stratified (data not shown), hence more exposed to excess of damaging sunlight.".

3. Line 591: MMA is the most volatile, you have to acknowledge this and that one important reason for not detecting it is the loss to the air. Unless you have carefully investigated the recovery as mentioned above.

Authors: We agree with the reviewer. The sentence has been improved: "Finally, regarding dMMA, the labile and volatile nature of this compound suggests that bacteria efficiently remineralize dMMA into ammonium (Lidbury et al., 2015b), and that MMA volatilizes quickly to the atmosphere, both processes contributing to the rapid depletion of MMA in surface waters." We assume that there are no analytical problems.

4. The same comment goes for lines 624-625: equilibration with air is the likely explanation along with recycling.

Authors: We agree with the reviewer and we have improved the statement accordingly: "This recycling of amines may explain their nanomolar concentrations in seawater, suggesting they may serve as valuable organic N sources for both phytoplankton and bacteria. Given their volatile nature, alkylamines may also be lost to the atmosphere."

**Technical corrections**

1. The name of the expedition is sometimes written as "PolarChange" and sometimes as "Polar Change"

Authors: PolarChange is the correct name of the expedition, we have fixed it in the revised manuscript.

2. Line 86: "a few studies" are mentioned but then only one is cited, from one of the co-authors of the present work. I'm curious if other studies were conducted with other methods and if you compared your findings. If non-existent, state so.

Authors: We have corrected the references. Since Fitzsimons et al. (2023) is a review, we believe it is sufficient as many references and amine measurements and concentrations are shown. Additionally, Fitzsimons et al. (2024) recently published another review on amines in the marine system.

3. Lines 166-172: please rewrite this paragraph so it's clear that the CPA and NaOH were added after the filters were in the autosampler vials.

Authors: We have rewritten this paragraph for better clarity.

4. Line 224: Please provide a reference for the Cryptophytes classification.

Authors: We have rephrased it to "Phytoplankton cells were detected with a 488 nm laser beam from their signatures in a plot of side scatter (SSC) *versus* red fluorescence (FL3), separating the picophytoplankton size class of 1–2 µm (sphere equivalent diameter, SED), and the nanophytoplankton size classes with SEDs of 2–7 µm, 7–15 µm, and 15–20 µm (Fig. S4). Within the nanophytoplankton, Cryptophytes (*Cryptomonas* spp.) were identified by their phycoerythrin signal in the FL3 vs orange fluorescence (FL2) plots (Marie et al., 2014)."

Marie et al., 2014: https://doi.org/10.1002/cyto.a.22517.

5. Line 272-273: provide a reference or a vendor for the software.

Authors: There is no commercial vendor for the software, as it was developed by M. Gorbunov, who has already been cited in the manuscript.

6. Line 279: clarify RT is room temperature the first time it appears in the text.

Authors: We have added "room temperature" as it appears the first time in the text.

7. Line 373: values are rounded differently here and in Table S1.

Authors: We have corrected the revised manuscript.

8. Table S1: check rounding, there is no "0.0" uncertainty.

Authors: We have added a number.

9.  Line 389-390: Please be consistent with chlorophyll units throughout the text.

Authors: Sure, we have corrected it.

10. Line 435: I believe you meant to cite Table S3 here instead of Fig S5.

Authors: Yes thanks, we have corrected it in the revised manuscript.